# Homophilic ATP1A1 binding induces activin A secretion to promote EMT of tumor cells and myofibroblast activation

Yi-Ing Chen [1], Chin-Chun Chang[1,2], Min-Fen Hsu [1], Yung-Ming Jeng[3], Yu-Wen Tien [4], Ming-Chu Chang[5,6], Yu-Ting Chang[5,6], Chun-Mei Hu [1 ✉] & Wen-Hwa Lee [1,7,8 ✉]

Tumor cells with diverse phenotypes and biological behaviors are influenced by stromal cells through secretory factors or direct cell-cell contact. Pancreatic ductal adenocarcinoma (PDAC) is characterized by extensive desmoplasia with fibroblasts as the major cell type. In the present study, we observe enrichment of myofibroblasts in a juxta-tumoral position with tumor cells undergoing epithelial-mesenchymal transition (EMT) that facilitates invasion and correlates with a worse clinical prognosis in PDAC patients. Direct cell-cell contacts forming heterocellular aggregates between fibroblasts and tumor cells are detected in primary pancreatic tumors and circulating tumor microemboli (CTM). Mechanistically, ATP1A1 overexpressed in tumor cells binds to and reorganizes ATP1A1 of fibroblasts that induces calcium oscillations, NF-κB activation, and activin A secretion. Silencing ATP1A1 expression or neutralizing activin A secretion suppress tumor invasion and colonization. Taken together, these results elucidate the direct interplay between tumor cells and bound fibroblasts in PDAC progression, thereby providing potential therapeutic opportunities for inhibiting metastasis by interfering with these cell-cell interactions.

[1] Genomics Research Center, Academia Sinica, Taipei, Taiwan. [2] Master Program for Cancer Biology and Drug Discovery, China Medical University and Academia Sinica, Taichung, Taiwan. [3] Department of Pathology, National Taiwan University Hospital, Graduate Institute of Pathology, College of Medicine, National Taiwan University, Taipei, Taiwan. [4] Department of Surgery, National Taiwan University Hospital, College of Medicine, National Taiwan University, Taipei, Taiwan. [5] Department of Internal Medicine, College of Medicine, National Taiwan University, Taipei, Taiwan. [6] Department of Internal Medicine, National Taiwan University Hospital, Taipei, Taiwan. [7] Drug Development Center, China Medical University, Taichung, Taiwan. [8] Department of Biological Chemistry, University of California, Irvine, USA. ✉email: CMHU1220@sinica.edu.tw; whlee@uci.edu

Tumor heterogeneity is the most distinct characteristic of cancer. Genetic instability in cancer cells and the influence of stromal cells in the tumor microenvironment are the two major drivers of tumor heterogeneity. These two forces propel tumor progression, metastasis, and treatment resistance. Numerous studies have shown that genetic instability of cancer cells generates tumor heterogeneity[1]. Genomic analysis of distinct tumor regions identified differential mutation profiles and gene expression patterns[2]. Epigenetic modifications activate aberrant transcription factors to regulate reversible transformation and de-differentiation that result in stem cell-like features in tumor cells[3]. In contrast, relatively few studies have explored how stromal cells affect tumor cell spatial heterogeneity[4]. This is a compelling question as the high genetic and transcriptional diversity found in stromal cells is known to contribute to tumor heterogeneity through cell-cell interactions occurring within the tumor microenvironment[5].

In the beginning of tumorigenesis when phenotypic normal epithelial cells encumber with mutations, differentiated cues derived from surrounding stromal cells would restrain malignant growth[6]. In the later stage of tumor progression, tumor-derived cytokines and growth factors out-weight tissue derived signals and actively reconstruct their microenvironment to form a supportive niche for tumorigenesis and metastasis. Tumor cells guide assembly of abnormal blood vessels to access oxygen and nutrients[7], recruit protumor and suppress antitumor immune cells[8], and educate fibroblasts for extracellular matrix remodeling and growth factor secretion[9]. Reciprocally, signals from stromal cells (including tumor-associated macrophages (TAMs) and fibroblasts) increase tumor heterogeneity by triggering epithelial-mesenchymal trans-differentiation or retro-differentiation of tumor cells resulting in the generation of cancer stem cells[10,11]. The finding that EMT occurs in the peripheral regions of tumor, which is adjacent to tumor stroma, may explain the common spatial heterogeneity seen in solid tumors[11,12]. However, a detailed mechanism of how tumors drive microenvironment remodeling remains elusive.

PDAC is characterized by a desmoplastic microenvironment with fibroblasts as the major cellular component[8]. Cancer-associated fibroblasts (CAFs) are a heterogeneous population that exhibit tumor-promoting or restraining functions[13,14]. For education of tumor-promoting CAFs, pancreatic cancer cells secrete growth factors, cytokines, and exosomes to stimulate fibroblast activation[15]. In turn, activated fibroblasts produce extracellular matrix, growth factors, and inflammatory cytokines to promote tumor proliferation, invasion, stemness features, chemoresistance, and immunosuppression[15]. Distinct from paracrine signaling, several observations implicate the biological importance of physical contact between tumor cells and fibroblasts. Lung cancer cells in contact with fibroblasts enhance tumor migration[16]. Tumor cell-fibroblast multicellular aggregates facilitate peritoneal adhesion and invasion of ovarian cancer cells for peritoneal metastasis[17]. The presence of fibroblast-containing tumor clusters in the bloodstream is associated with clinical metastasis in breast cancer[18], and generates resistance to shear stress in cases of prostate cancer[19]. These observations suggest tumor-promoting roles for direct tumor-fibroblast contact. However, how this crosstalk between tumor cells and fibroblasts is mediated remains an important question.

In this communication, we demonstrate direct contact between fibroblasts and tumor cells in the invasion front of the primary tumor and in CTM during metastasis. These juxta-tumoral fibroblasts serve as a stromal niche to support EMT of tumor cells in the tumor invasion front and blood circulation, leading to poor clinical outcomes for patients. Furthermore, our results show that direct contact between these two cell types is in part mediated by homophilic ATP1A1 binding. Through this interaction, activin A secreted from NF-κB activated fibroblasts induces EMT tumor cells and αSMA$^+$ fibroblasts along the tumor-stroma interface. Elucidating the mechanistic basis and subsequent effects of this interaction between fibroblasts and PDAC tumor cells may provide novel opportunities to disrupt this crosstalk and improve PDAC management.

## Results

### The presence of a prominent subpopulation of αSMA$^+$ fibroblasts surrounding tumor cell nests correlates with a worse clinical prognosis.

To explore the role of fibroblasts in juxta-tumoral regions, we used an antibody against αSMA, which is a biomarker for myofibroblasts[20], to immunostain pancreatic tumor specimens derived from both mice and human patients. In Pdx1-Cre; LSL-Kras$^{G12D/+}$; Trp53$^{flox/flox}$ (PdKP53) mouse models of PDAC, αSMA$^+$ fibroblasts were preferentially located in juxta-tumoral regions, as defined by the expression levels of αSMA within 0–10 μm versus 10–20 μm distance from tumor margins (Fig. 1a–c). In contrast, when staining all fibroblasts with vimentin, a biomarker of mesenchymal cells, there was no difference in expression when comparing 0–10 μm versus 10–20 μm distances from tumor margins (Fig. 1a, b, d). A similar spectrum of peritumoral localization of αSMA$^+$ fibroblasts was observed in PDAC patient specimens. αSMA$^+$ fibroblasts were enriched in juxta-tumoral regions (Fig. 1e–g), while VIM$^+$ fibroblasts were evenly distributed throughout the tumor stroma (Fig. 1e, f, h). To examine the clinical significance of juxta-tumoral location of αSMA$^+$ fibroblasts, a retrospective cohort containing stage II PDAC patients ($n = 85$) who underwent pancreaticoduodenectomy at National Taiwan University Hospital (NTUH) from 2007 to 2015 was studied. Patient characteristics are listed in Table 1. We then used receiver operating characteristic curve (ROC curve) analysis to set the cut-off value for high versus low αSMA expression levels, and performed a Kaplan–Meier survival analysis to compare patient prognosis according to juxta-tumoral αSMA expression. The results show that patients with high juxta-tumoral αSMA expression (0–10 μm) had shorter progression-free survival ($p = 0.0067$) and overall survival ($p = 0.0056$) compared to patients with low juxta-tumoral αSMA expression (Fig. 1i, k), while αSMA expression levels in the regions of 10–20 μm distant to tumor cells were not correlated with patient outcome (Figs. 1j, l). These results suggest that enhanced tumor-fibroblast interaction, as indicated by the enrichment of activated myofibroblasts in peritumoral regions, may promote elevated malignancy in pancreatic cancers.

### αSMA$^+$ fibroblasts are directly associated with cancer cells.

To test whether these juxta-tumoral myofibroblasts were directly attached to tumor cells, pancreases from Pdx1-Cre; LSL-Kras$^{G12D/+}$; Trp53$^{flox/flox}$ (PdKP53) mice and control littermates (Pdx1-Cre; LSL-Kras$^{+/+}$; Trp53$^{flox/flox}$ (WT)) between 6 to 7 weeks old were collected and mechanically dissociated to cell clusters with a gentle MACS$^{TM}$ dissociator (Milteny Biotec) (Fig. 2a). At this stage, pancreatic tumors about $0.3 \times 0.3 \times 0.3$ cm in size were frequently observed in PdKP53 mice, while no tumor was observed in WT littermate mice (Supplementary Fig. 1a). These cell clusters were stained with antibodies against EpCAM (epithelial marker) and PDGFRα (fibroblast marker), and the EpCAM$^+$/PDGFRα$^+$ cell clusters were isolated by fluorescence-activated cell sorting (FACS) and further analyzed by immunostaining with anti-CK19 (PDAC marker) and anti-αSMA (myofibroblast marker) antibodies (Fig. 2a). Cell clusters with CK19- and αSMA-positive staining were observed in primary PDAC tumors (Fig. 2b). In addition, circulating tumor micro-emboli (CTM) were isolated from the peripheral blood of

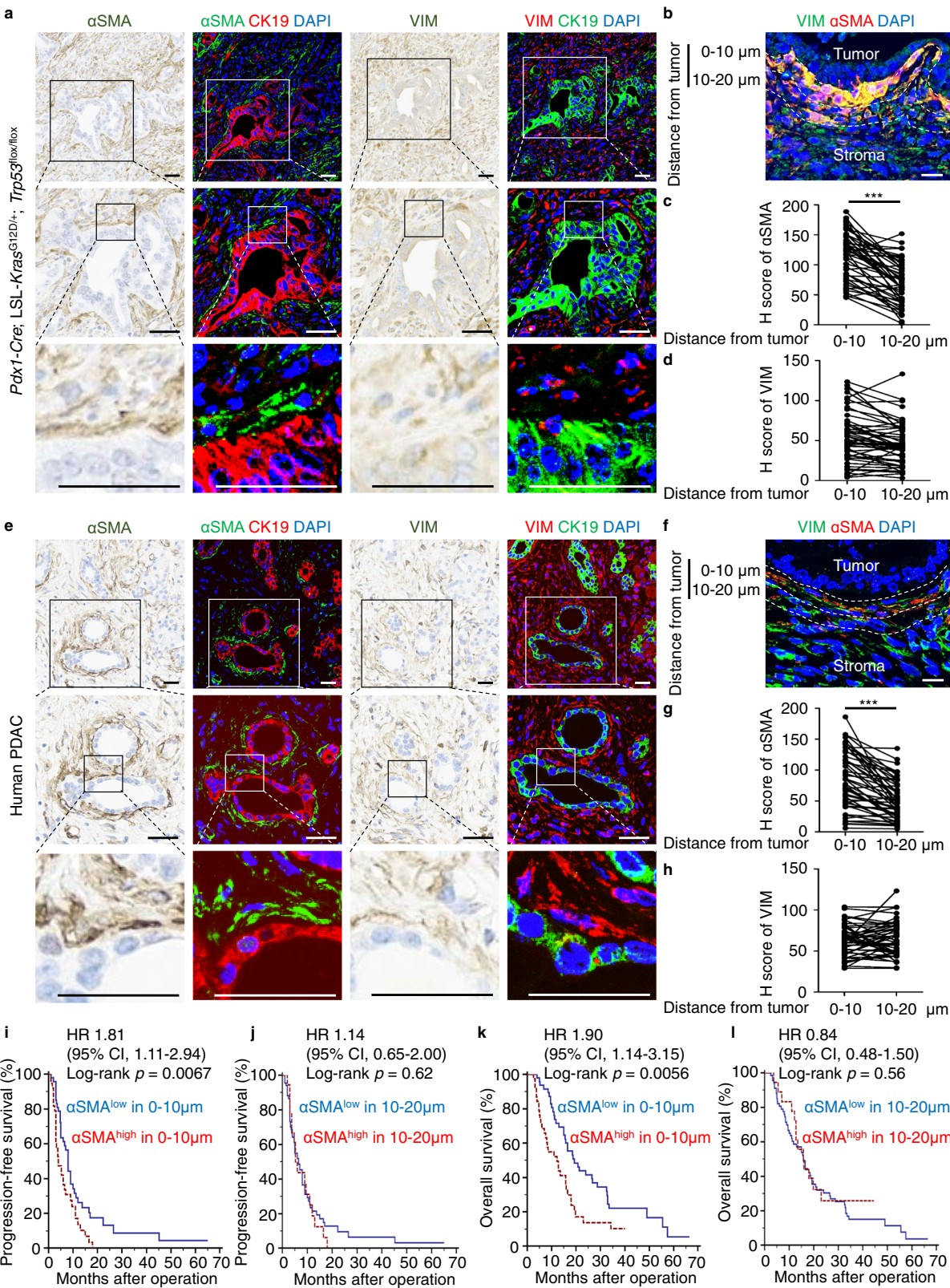

PdKP53 mice and PDAC patients (Supplementary Fig. 1b) using Accuspin tubes (Merck) and Cytospin (Supplementary Fig. 1c). These CTMs contained cell clusters with CK19$^+$/αSMA$^+$ and CK19$^+$/CD45$^+$/αSMA$^+$ biomarkers when immunostained with antibodies against CK19, αSMA, and CD45 (immune cell marker) (Fig. 2c, d). These results suggest that tumor cells and fibroblasts are directly associated with physiological conditions.

To establish a tissue culture model for investigating tumor cell and fibroblast interaction, we used a simplified 3D-Matrigel culture assay[21]. Pancreatic fibroblasts (mPSC-RFP and PSC-GFP) and tumor cells (PdKP53-GFP, KPC-GFP, BxPC-3-RFP, and SU.86.86-RFP) were mono-cultured or co-cultured in Matrigel (Fig. 2e, h). After 24 hours of mono-culture, fibroblasts or tumor cells formed homocellular aggregates (Fig. 2f, g, i, j; Supplementary Fig. 1d, e).

**Fig. 1 The presence of a prominent subpopulation of αSMA$^+$ fibroblasts surrounding cancer cell nests is associated with poor prognosis.**
**a** Representative IHC images of αSMA/CK19/DAPI or VIM/CK19/DAPI expression in the 6 to 7 weeks old *Pdx1-Cre*; LSL-*Kras*$^{G12D/+}$; *Trp53*$^{flox/flox}$ (PdKP53) transgenic mice. Scale bar, 50 μm. **b** Representative IHC images of VIM/αSMA/DAPI expression in regions with 0–10 μm and 10–20 μm distance from tumor-stromal interface in PdKP53 transgenic mice. Scale bar, 10 μm. **c** H score of αSMA expression levels in regions with 0–10 μm and 10–20 μm distance from tumor-stromal interface in PdKP53 transgenic mice (*n* = 50). ***p* < 0.001 (two-tailed t-test). **d** H score of VIM expression levels in regions with 0–10 μm and 10–20 μm distance from tumor-stromal interface in PdKP53 transgenic mice (*n* = 50). **e** Representative IHC images of αSMA/CK19/DAPI or VIM/CK19/DAPI expression in human PDAC specimens. Scale bar, 50 μm. **f** Representative IHC images of VIM/αSMA/DAPI expression in regions with 0–10 μm and 10–20 μm distance from tumor-stromal interface in human PDAC specimens. Scale bar, 10 μm. **g** H score of αSMA expression levels in regions with 0–10 μm and 10–20 μm distance from tumor-stromal interface in human PDAC specimens (*n* = 50). ***p* < 0.001 (two-tailed t-test). **h** H score of VIM expression levels in regions with 0–10 μm and 10–20 μm distance from tumor-stromal interface in human PDAC specimens (*n* = 50). **i**, **k** Kaplan–Meier analysis of progression-free survival **i** and overall survival **k** according to H score of αSMA expression levels in regions within 10 μm distance from tumor (*p* = 0.0067 and *p* = 0.0056, separately, Log-rank test). Blue line, αSMA$^{low}$ in 0–10 μm, *n* = 47; Red line, αSMA$^{high}$ in 0–10 μm, *n* = 38. **j**, **l** Kaplan–Meier analysis of progression-free survival **j** and overall survival **l** according to H score of αSMA expression levels in regions within 10–20 μm distance from tumor. Blue line, αSMA$^{low}$ in 10–20 μm, *n* = 67; Red line, αSMA$^{high}$ in 10–20 μm, *n* = 18.

| Table 1 Clinical characteristics of PDAC patients (*n* = 85). | |
|---|---|
| **Characteristics** | **No.** |
| Age | |
| Mean (year, SD) | 64.9 (12.4) |
| Range | 26–88 |
| <60 | 32 |
| ≧60 | 53 |
| Gender | |
| Male | 49 |
| Female | 36 |
| AJCC stage | |
| IIA | 32 |
| IIB | 52 |
| Unknown | 1 |
| Tumor grade | |
| G1 | 9 |
| G2–G3 | 76 |
| Primary tumor extent (T) | |
| T2 | 7 |
| T3 | 78 |
| Lymph node status (N) | |
| N0 | 31 |
| N1 | 51 |
| Unknown | 3 |
| Distant metastasis (M) | |
| M0 | 85 |
| Resection margin status (R) | |
| R0 | 65 |
| R1 | 19 |
| Unknown | 1 |
| Lymphovascular invasion | |
| Negative | 23 |
| Positive | 54 |
| Unknown | 8 |
| Perineural invasion | |
| Negative | 8 |
| Positive | 70 |
| Unknown | 7 |

Co-culture of tumor cells and fibroblasts resulted in large heterocellular aggregates (Fig. 2f, g, i, j; Supplementary Fig. 1d, e). Fibroblasts that form aggregates with tumor cells express αSMA (Supplementary Fig. 1f–h). These results suggest that a direct cell-cell contact between tumor cells and myofibroblasts exists.

**Direct contact with fibroblasts promotes tumor malignancy by enhancing epithelial-mesenchymal transition.** To decipher the phenotypic effects of direct contact between tumor cells and fibroblasts, we compared sphere formation and collective invasion among tumor cells in mono-cultured, indirect co-cultured, or direct co-cultured conditions. For the sphere-formation assay, pancreatic tumor cells (BxPC-3-RFP, SU.86.86-RFP, PdKP53-GFP, and KPC-GFP) were mono-cultured, indirect (transwell) co-cultured (ID), or direct co-cultured (D) with pancreatic fibroblasts (PSC-GFP and mPSC-RFP) in ultra-low attachment plates. In comparison to mono-cultured or indirect (transwell) co-cultured conditions, tumor cells in direct contact with fibroblasts significantly increased the number of tumor spheroids (Fig. 3a, b; Supplementary Fig. 2a, b). Moreover, positive correlations between the number of fibroblasts and tumor spheroids in direct-contact conditions revealed the role of fibroblasts in promoting self-renewal and proliferation of tumor cells in an anchorage-independent manner (Fig. 3a, b; Supplementary Fig. 2a, b). For 3D spheroid models of collective cancer cell invasion, human pancreatic tumor spheroids (BxPC-3-RFP spheroids) were mono-cultured, indirect co-cultured (ID), or direct co-cultured (D) with human pancreatic fibroblasts (PSC-GFP) in Matrigel (Fig. 3c). Enhancement of collective tumor invasion was noted after direct contact with fibroblasts (Fig. 3d–f).

To explore the mechanism of direct tumor-fibroblast contact-mediated tumor malignancy, RNA-seq analysis was performed to compare gene expression profiles of tumor cells among mono-, indirect-, or direct- co-cultured conditions. We identified 464 common up-regulated (≥ 2-fold) genes in BxPC-3 and SU.86.86 cells upon direct contact with fibroblasts (Fig. 3g). Using the Ingenuity Pathway Analysis (IPA) system, the top canonical pathways associated with these genes were related to fibroblast activation and regulation of EMT (Fig. 3h). Since tumor sphere formation and invasion were highly associated with the biological properties of EMT[22,23], we examined the expression of EMT signature genes in tumor cells directly bound to fibroblasts. Indeed, mRNA and protein expression levels of an epithelial gene (CLDN1) were decreased, while those of mesenchymal (VIM) and EMT regulator (Snail, Twist1, ZEB1, and ZEB2) genes were increased in tumor cells (Fig. 3i, j). Furthermore, up-regulation of EMT in tumor cells adjacent to fibroblasts was observed when we used tumor xenografts co-injected with PSCs (Supplementary Fig. 2c, d). Since EMT plasticity enhances tumor progression and metastasis[24], we then compared the tumorigenic and colonization abilities between tumor cells and tumor-fibroblast clusters and found that tumor-fibroblast clusters significantly increased tumor growth and liver colonization activities (Fig. 3k–n). Thus, fibroblasts at the tumor-stromal interface provide a niche for adjacent tumor cells to gain EMT properties for further progression.

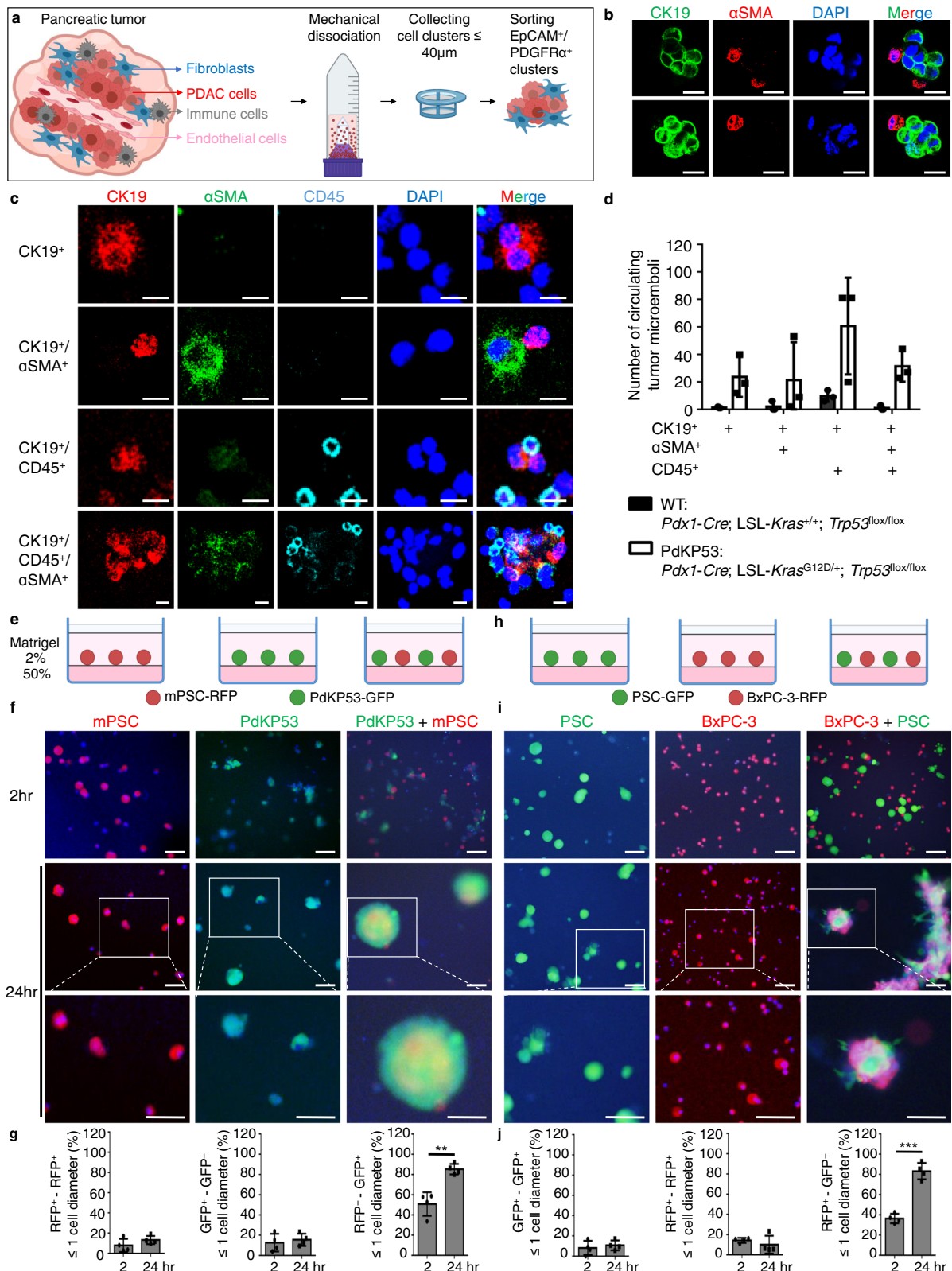

**Direct tumor-fibroblast contact triggers activin A secretion leading to EMT of tumor cells and myofibroblast activation.** To gain mechanistic insight into direct contact-mediated EMT of tumor cells, we employed a cytokine array (Quantibody Human Cytokine Array 4000; RayBiotech) to measure cytokine secretion during co-culturing of BxPC-3 cells with fibroblasts (PSCs) in transwell (indirect, ID)- and direct (D) configurations. Activin A

was the only cytokine that had a ≥ 2-fold increment difference upon direct co-culture with fibroblasts (Fig. 4a). It was noted that activin A signaling is important for the induction and maintenance of mesenchymal phenotypes[25]. We then treated tumor cells with activin A recombinant protein and observed an enhancement of EMT gene expression and tumor sphere formation (Fig. 4b–d; Supplementary Fig. 3a). Conversely, treatment

**Fig. 2 αSMA⁺ fibroblasts directly bind to tumor cells. a** The schematic procedure of cell cluster isolation. Pancreases were isolated from 6 to 7 weeks old PdKP53 transgenic mice. The whole pancreas was cut into pieces of <1 mm³ and underwent mechanical dissociation to cell clusters <40 μm. EpCAM⁺/PDGFRα⁺ cell clusters were captured by FACS. **b** Representative immunofluorescence (IF) images of CK19/αSMA/DAPI expression in EpCAM⁺/PDGFRα⁺ clusters. Scale bar, 10 μm. **c** Representative IF images of CK19/αSMA/CD45/DAPI expression in circulating tumor microemboli (CTM) in 6 to 7 weeks old PdKP53 transgenic mice. Scale bar, 10 μm. **d** Number of CTM in 6 to 7 weeks old *Pdx1-Cre*; LSL-*Kras*⁺/⁺; *Trp53*ᶠˡᵒˣ/ᶠˡᵒˣ (WT) and PdKP53 mice. Each sample was pooled by three mice. Thus, total of 9 PdKP53 mice were used for triplicate data. Values were presented as mean ± SD ($n = 3$). **e–g** 3D-Matrigel mono-culture or co-culture assay of mouse pancreatic stellate cells labeled with RFP (mPSC-RFP) and mouse pancreatic cancer cells labeled with GFP (PdKP53-GFP). The culture condition **e** and representative images **f**. Scale bar, 100 μm. **g** After 2 h and 24 h, the percentage of RFP⁺-RFP⁺, GFP⁺-GFP⁺, or RFP⁺-GFP⁺ cells with distance ≤ 1 cell diameter were determined by Imaris software at indicated time points. Values were presented as mean ± SD ($n = 4$). **h–j** 3D-Matrigel mono-culture or co-culture assay of human pancreatic stellate cells labeled with GFP (PSC-GFP) and human pancreatic cancer cells labeled with RFP (BxPC-3-RFP). The cultural condition **h** and representative images **i**. Scale bar, 100 μm. **j** After 2 h and 24 h, the percentage of RFP⁺-RFP⁺, GFP⁺-GFP⁺, or RFP⁺-GFP⁺ cells with distance ≤ 1 cell diameter were determined by Imaris software at indicated time points. Values were presented as mean ± SD ($n = 4$). **$p < 0.01$, ***$p < 0.001$ (two-tailed t-test). Panel **a**, **e**, **h** were created with BioRender.com.

with anti-activin A antibody reversed the EMT state, as indicated by the down-regulation of VIM, Snail, and ZEB1 (Fig. 4e) and significantly decreased sphere formation as well as 3D spheroid invasion upon direct contact with fibroblasts (Fig. 4f–h; Supplementary Fig. 3b). To test whether activin A plays a role in fibroblast-promoted tumorigenesis in vivo, we injected anti-activin A antibody or follistatin (an activin antagonist) every three days for a total of four injections into NOD/SCID/IL2Rγᴺᵘˡˡ (NSG) mice xenografted with BxPC-3 tumor cells and human PSCs. Co-injection of human PSCs enhanced tumorigenicity significantly (Fig. 4i–l), while neutralization of activin A with anti-activin A antibody or follistatin significantly decreased tumor growth only in cases where PSCs were also injected (Fig. 4i–l). This result suggests that activin A has a critical role in fibroblast-mediated tumor progression.

It was noted that activin A promoted a pro-fibrotic gene expression signature in skin fibroblasts[26] and myofibroblast differentiation in endometrial mesenchymal stem cells[27]. Since αSMA⁺ myofibroblasts were enriched in the juxta-tumoral region (Fig. 1a–c, e–g) and adjacent to tumor cells with EMT (Supplementary Fig. 2c, d), activin A secreted from direct tumor-fibroblast contact may enhance myofibroblast activation. To test this hypothesis, we treated fibroblasts with activin A recombinant protein and found that both mRNA and protein levels of αSMA and collagen I were up-regulated (Fig. 4m, n) and contractile forces of fibroblasts were increased (Supplementary Fig. 3c). Consistently, αSMA, collagen I, and collagen contractility were also increased in fibroblasts in direct contact with tumor cells (Fig. 4o, p; Supplementary Fig. 3d), while neutralization with anti-activin A antibody diminished αSMA expression and collagen contractility in fibroblasts (Fig. 4q; Supplementary Fig. 3e). Interestingly, when we measured the amount of αSMA⁺ cells after neutralization with anti-activin A antibody or treatment with follistatin in tumor xenografts co-injected with PSCs in mice, we found that the number of αSMA⁺ cells adjacent to the tumor was significantly reduced in these mice (Fig. 4r–u). Taken together, these results suggest that activin A secreted during direct contact between fibroblasts and tumor cells has two functions: enhancement of EMT in tumor cells and promotion of myofibroblast differentiation.

**Activin A is secreted from fibroblasts primarily upon direct contact with tumor cells**. To address the origin of activin A-producing cells, we used the protein transport inhibitor, monensin, to trap cytokines in the Golgi complex for immunofluorescence staining[28]. Activin A was highly expressed in fibroblasts directly bound to tumor cells (Fig. 5a, b). To distinguish whether activin A production is a primary effect of direct contact with tumor cells or a secondary effect induced by feedback

signals from tumor cells, we treated tumor cells with 2% paraformaldehyde (PFA)-fixed fibroblasts or reciprocally treated fibroblasts with 2% PFA-fixed tumor cells. Fibroblasts secreted activin A only upon direct contact with tumor membrane proteins, but not the reverse (Fig. 5c). Interestingly, not all the tumor cell membranes induced activin A secretion to the same level. Cell membranes from MIA PaCa-2 and PANC-1 cell lines had little or no effect. To further confirm the origin of activin A-producing cells in human PDAC tumor specimens, mRNAs of activin A (INHBA) and αSMA (ACTA2) were co-stained as individual spots by in situ mRNA hybridization using RNAscope (ACDbio). Indeed, the majority of activin A was expressed in αSMA⁺ fibroblasts in tumor-adjacent regions (Fig. 5d, e). Moreover, EMT occurred in tumor cells adjacent to activin A-secreting fibroblasts (Fig. 5f, g). These results suggest that activin A secreted from fibroblasts upon direct contact with tumor cells promotes EMT of tumor cells. To examine the potential clinical significance of the number of activin A-secreting fibroblasts adjacent to tumor cells in PDAC, we performed Kaplan–Meier survival analysis and found that a higher number of activin A-secreting cells (INHBA, 2–3 +) in stromal regions within 0–10 μm from tumor margins was significantly correlated with shorter overall survival ($p = 0.01$) compared to a lower number of activin A-secreting cells (INHBA, 0–1 +) (Fig. 5h, i). These results suggest that activin A-secreting fibroblasts adjacent to tumor cells play a critical role in cancer progression.

**ATP1A1 of tumor cells is the mediator of contact-induced activin A secretion**. To address the mechanism of how direct contact between tumor cells and fibroblasts induces activin A secretion, we first examined the key elements of this contact. We showed previously (Fig. 5c) that tumor membrane proteins found on several tumor cell lines were able to induce fibroblast activin A production, though this was not true for all the cell lines tested. Interestingly, two different sources of fibroblasts secreted activin A in response to purified tumor membranes in a similar manner (Fig. 6a), suggesting that activin A secretion is a shared responsibility of fibroblasts to tumor cells. Based on these results, we prepared tumor membrane proteins from cell lines with high (BxPC3/SU.86.86) or low (MIA PaCa-2/PANC-1) activin A-inducing activity to screen for which membrane protein(s) trigger contact-dependent activin A secretion. We added Sulfo-SBED, a biotin-labeled cross-linker with an amine reactive group and a UV-activated aryl azide (Supplementary Fig. 4a), to prepared membrane proteins from BxPC-3, SU.86.86, MIA PaCa-2, and PANC-1 cells. This resulted in cross-linking of the Sulfo-SBED to the -NH2 moiety of the proteins. The Unbound cross-linker was removed, and the Sulfo-SBED-bound membrane proteins were then added to fibroblasts for 5 h. The photoreactive site of Sulfo-SBED was then

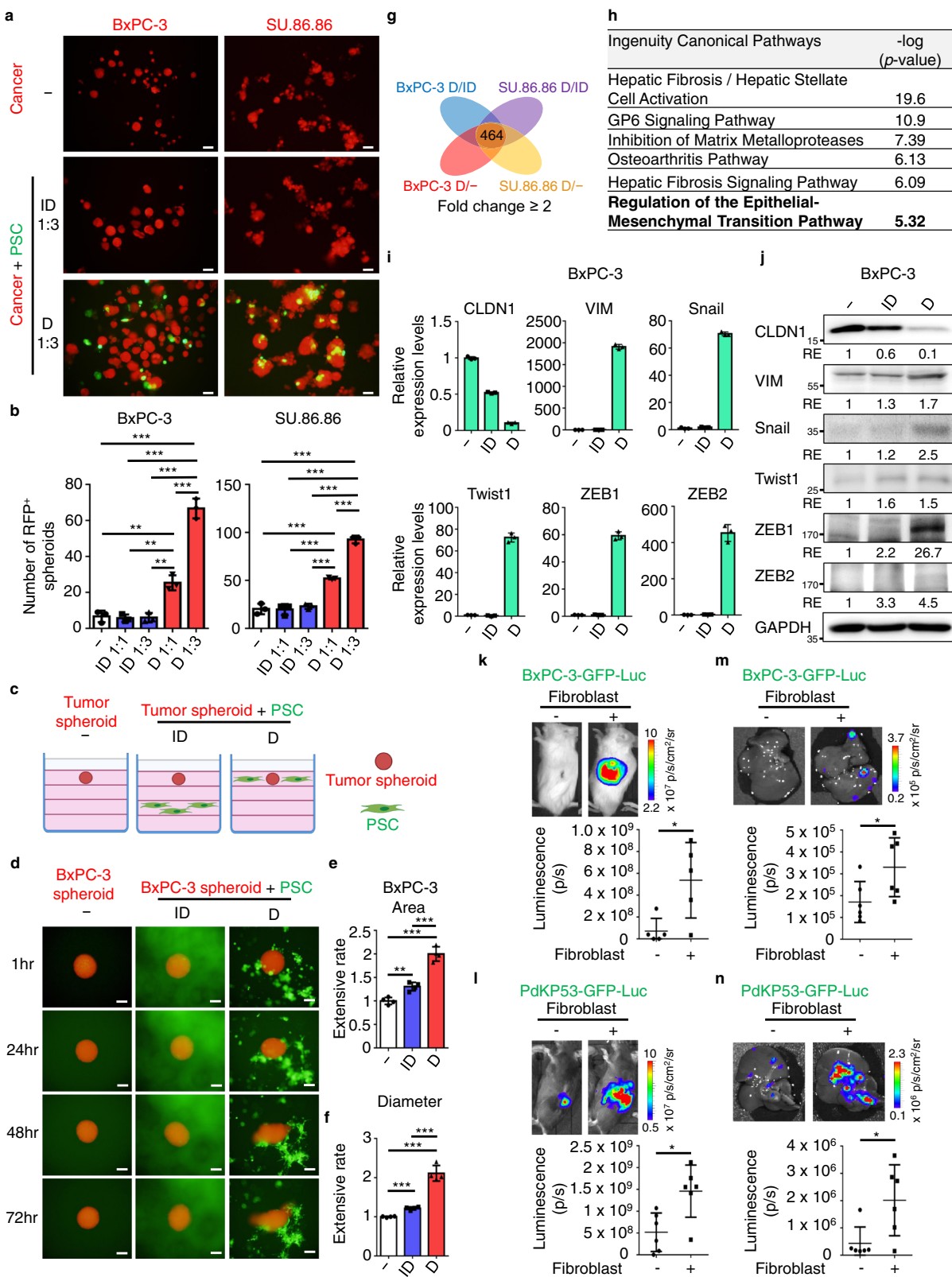

activated with UV to cross-link the adjacent proteins and the cross-linked proteins were purified by monomeric-avidin beads (Supplementary Fig 4a) for western blotting analysis. A 130 kDa band recognized by streptavidin-HRP was specifically detected in BxPC-3 and SU.86.86 samples (Fig. 6b). To enhance the purity of the proteins for mass spectrometry, cytosolic and nuclear proteins of fibroblasts were immobilized by intracellular hydrogelation[29,30] to

reduce contamination. The 130 kDa band identified in SDS-PAGE was eluted for mass spectrometry (Supplementary Fig. 4b) analysis, which revealed five plasma membrane proteins (ATP1A1, ATP1A2, ATP1A3, EMC1, and ITGB1) as potential candidates (Fig. 6c).

To validate which membrane protein was involved in contact-mediated activin A secretion, each of the five potential candidates

**Fig. 3 Direct contact with fibroblasts promotes EMT of tumor cells. a, b** Sphere-formation assay of mono-cultured, indirect co-cultured (ID), and direct co-cultured (D) tumor cells (BxPC-3-RFP and SU.86.86-RFP) with fibroblasts (PSC-GFP). Representative images **a** and quantification results **b** of RFP$^+$ spheroids. Scale bar, 100 μm. Values were presented as mean ± SD ($n = 3$). **c–f** 3D spheroid invasion assay of mono-cultured, indirect co-cultured (ID), and direct co-cultured (D) tumor spheroids (BxPC-3-RFP spheroids) with fibroblasts (PSC-GFP cells). The culture condition **c** and representative images **d**. Scale bar, 200 μm. The extensive rate of each tumor spheroid was calculated by normalizing RFP$^+$ area at 72 h to RFP$^+$ area at 1 h ($n = 4$). Panel **c** was created with BioRender.com. **e** and RFP$^+$ maximal diameter at 72 h to RFP$^+$ diameter at 1 h **f**. Values were presented as mean ± SD ($n = 4$). **g** The differential gene expression analysis with RNA-seq. Overlap of the commonly up-regulated genes (464) in tumor cells (BxPC-3-RFP and SU.86.86-RFP) after direct (D) co-cultured with fibroblasts compared to mono-cultured (−) or indirect co-cultured (ID) conditions were used for Ingenuity Pathway Analysis (IPA). **h** Top 6 canonical pathways with the largest -log ($p$-value) calculated by Fisher's Exact Test in IPA were shown. **i** Quantitative real-time PCR analyses of CLDN1, VIM, Snail, Twist1, ZEB1, and ZEB2 expression were performed to compare gene expression levels in mono-cultured, indirect co-cultured (ID), and direct co-cultured (D) tumor cells (BxPC-3-RFP) with fibroblasts. FACS was performed to collect RFP$^+$ tumor cells after 24 h of mono- or co-cultured conditions. Values were presented as mean ± SD ($n = 3$). **j** Western blotting analysis of proteins harvested from mono-cultured, indirect co-cultured (ID), and direct co-cultured (D) tumor cells (BxPC-3-RFP) with fibroblasts. FACS was performed to collect RFP$^+$ tumor cells after 48 h of mono- or co-cultured conditions. **k** BxPC-3-GFP-Luc cells with and without direct contact with fibroblasts were injected into the pancreas of NSG mice ($n = 5$). **l** PdKP53-GFP-Luc cells with and without direct contact with fibroblasts were injected into the pancreas of C57BL/6 mice ($n = 6$). **m, n** The effect of direct contact with fibroblasts on liver metastasis. **m** BxPC-3-GFP-Luc cells with and without direct contact with fibroblasts were injected into the spleen of NSG mice ($n = 6$). **n** PdKP53-GFP-Luc cells with and without direct contact with fibroblasts were injected into spleen of C57BL/6 mice ($n = 6$). Tumor growth and liver metastasis were assessed by IVIS bioluminescent analysis (Lumina, Perkin Elmer) after 2 weeks of orthotopic or splenic injection. The bioluminescent signal (pseudocolor) was recorded as photons per second (p/s). Values were presented as mean ± SD. *$p < 0.05$, **$p < 0.01$, ***$p < 0.001$ (two-tailed t-test).

were knocked down in BxPC-3 cells for subsequent contact-mediated activin A induction assays (Fig. 6d; Supplementary Fig. 4c). Knockdown of ATP1A1 and EMC1 significantly reduced contact-mediated activin A secretion (Fig. 6d), and the effect of ATP1A1 was more prominent than EMC1. This observation was repeated using a different set of knockdown clones (Supplementary Fig. 4d–g). Consistently, neutralization of ATP1A1 inhibited activin A secretion more significantly than that of EMC1 (Fig. 6e).

To further confirm that ATP1A1 plays a major role in mediating tumor progression, we performed direct contact-mediated sphere formation (Supplementary Fig. 5a–d), 3D spheroid invasion (Supplementary Fig. 5e, f), and liver colonization (Supplementary Fig. 5g, h) assays using tumor cells with shRNA-mediated ATP1A1 knockdown (compared to control shRNA). Tumor cells depleted of ATP1A1 demonstrated a reduction in all three activities. These results suggested that ATP1A1 on tumor cell membranes is important for inducing activin A secretion from fibroblasts and EMT of tumor cells. Interestingly, ATP1A1 is overexpressed in BxPC-3 and SU.86.86 cells, but not in MIA PaCa-2 and PANC-1 cells (Supplementary Fig. 5i). These results may explain why tumor cells overexpressing ATP1A1 can bind to fibroblasts and induce juxtacrine activin A production to promote EMT of tumor cells along with the tumor-stromal interface.

**Homophilic ATP1A1 interactions between tumor cells and fibroblasts trigger activin A secretion from fibroblasts.** To discover the binding partner of ATP1A1 in fibroblasts, ATP1A1-Flag-bound M2 agarose beads, control Flag-bound M2 agarose beads, or ATP1A1-Flag-bound HA agarose beads were incubated with biotin-labeled plasma membrane proteins from fibroblasts (Supplementary Fig. 6a). Potential interacting proteins of ATP1A1 were pulled-down by the agarose beads and released by detergent for further capturing by monomeric-avidin beads (Supplementary Fig. 6a). Western blotting was performed under reducing conditions and the binding partner of ATP1A1 in fibroblasts was visualized as a specific band in the ATP1A1-Flag-bound M2 agarose bead group by streptavidin-HRP (Fig. 6f, left panel). The same gel was re-probed with anti-Flag antibodies to exclude the potential contamination from ATP1A1-Flag of 293 T cells (Fig. 6f, right panel). A specific band around 130 kDa was cropped for mass spectrometry analysis (Supplementary Fig. 6b), which revealed ATP1A1 itself (Fig. 6g). To support this

finding, knockdown of ATP1A1 in fibroblasts significantly decreased direct contact-mediated activin A secretion and sphere formation (Fig. 6h; Supplementary Fig 6c–g).

To show that homophilic ATP1A1 interactions occur between tumor cells and fibroblasts, immunofluorescence staining of ATP1A1 was performed in co-cultured tumor cells (BxPC-3-RFP or SU.86.86-RFP cells) and fibroblasts (PSC-GFP). The positive signals of ATP1A1 were co-localized with RFP or GFP signals at the tumor-fibroblast interface (Fig. 6i). Therefore, tumor cells and fibroblasts both express ATP1A1 at the junctions between the two cell types. To further confirm this observation, we stably expressed ATP1A1-GFP in BxPC-3 cells and ATP1A1-RFP in PSCs. The respective GFP and RFP fluorescence signals were co-localized at the cell-cell junctions during the formation of live tumor cell-fibroblast contacts measured by SP8 confocal microscope (Leica Microsystems) (Fig. 6j). These results strongly suggest the presence of homophilic ATP1A1 interaction between tumor cells and fibroblasts.

To explore the potential significance of the above finding, we performed IHC staining with anti-ATP1A1 antibodies and RNAscope analysis for INHBA (activin A) using serial sections of human PDAC tumor specimens. ATP1A1 was enriched in basolateral membranes for tumor-tumor contacts and tumor-fibroblast contacts (Fig. 6k). INHBA was expressed by tumor-adjacent spindle-shaped cells and expression levels were positively correlated with those of ATP1A1 at the tumor-stromal interface (Fig. 6k, l). We conclude that homophilic ATP1A1 interactions between tumor cells and fibroblasts trigger activin A secretion from fibroblasts that promotes tumor progression.

**Homophilic ATP1A1 interactions reorganize ATP1A1 in the plasma membrane of fibroblasts and trigger intracellular Ca$^{2+}$ oscillations that stimulate NF-κB signaling and activin A secretion.** To explore how the homophilic ATP1A1 interactions induce activin A production, we analyzed the amounts of, and signal transduction associated with, ATP1A1 in both fibroblasts and tumor cells. We used intracellular hydrogelation procedures[29,30] to enrich plasma membrane fractions of BxPC-3 cells and PSCs (see Methods) and found that the amount of plasma membrane-localized ATP1A1 in fibroblasts, but not in tumor cells, was specifically increased upon direct tumor-fibroblast contact (Fig. 7a). Consistently, results of immunofluorescence staining without membrane permeabilization also showed enrichment of plasma membrane-localized ATP1A1 in

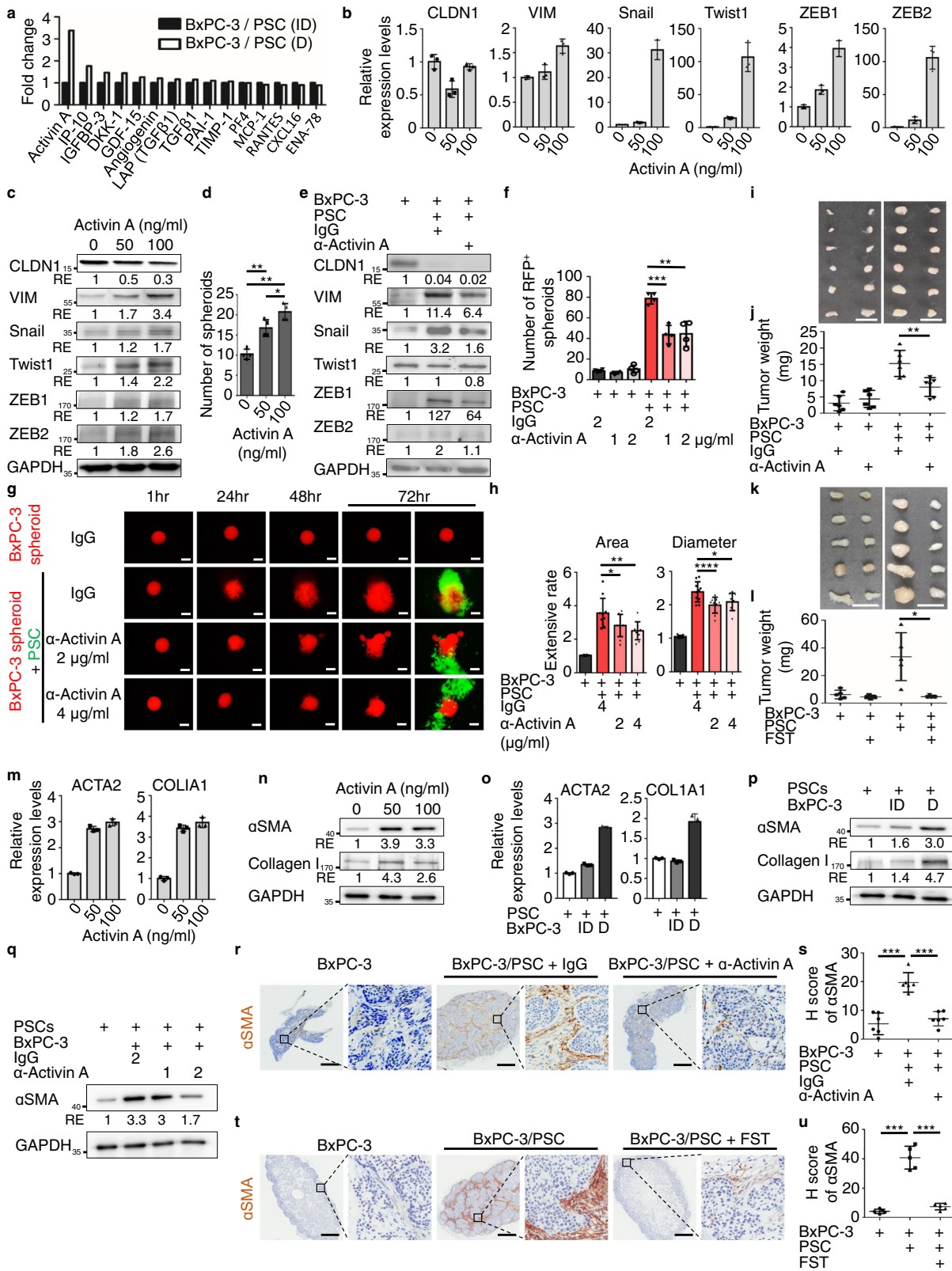

fibroblasts upon direct contact (Fig. 7b, c). Interestingly, the increased membrane-localized ATP1A1 in fibroblasts during direct contact relied on high ATP1A1 expression in tumor cells, but not in ATP1A1-knockdown tumor cells (Fig. 7d, e). These results indicate that fibroblast ATP1A1 is trafficked to the adhesion junctions of fibroblasts upon contact with high levels of ATP1A1 from tumor cells, indicating a re-organization of the fibroblast membrane-

localized ATP1A1. This re-organization may strengthen homophilic ATP1A1 interaction between tumor cells and fibroblasts.

Next, we investigated the ATP1A1-associated signaling responsible for driving activin A production. RNA sequencing data showed that the primary signaling pathways activated in fibroblasts upon direct contact with ATP1A1-overexpressing tumor cells (BxPC-3 and SU.86.86 cells, but not MIA PaCa-2 and

**Fig. 4 Direct tumor-fibroblast contact triggers activin A secretion leading to EMT of tumor cells and myofibroblast activation. a** Human cytokine array used conditional medium from transwell (indirect, ID) versus direct (D) co-culture of BxPC-3 with pancreatic stellate cells (PSCs). Cytokine levels in direct co-cultured condition were normalized to indirect co-cultured condition. **b** Quantitative real-time PCR analyses of CLDN1, VIM, Snail, Twist1, ZEB1, and ZEB2 expression were performed in BxPC-3 cells treated with recombinant human Activin A (50 and 100 ng/ml) compared to untreated cells. Values were presented as mean ± SD ($n = 3$). **c** Western blotting analysis of proteins harvested from BxPC-3 cells treated with recombinant human activin A (50 and 100 ng/ml) compared to untreated cells. **d** Sphere-formation assay of BxPC-3 cells treated with recombinant human activin A compared to untreated cells. Values were presented as mean ± SD ($n = 4$). **e** Western blotting analysis of proteins harvested from mono-cultured and direct co-cultured (D) tumor cells (BxPC-3-RFP) treated with IgG (1 μg/ml) or anti-activin A-neutralizing antibody (1 μg/ml, 693604, Biolegend). FACS was performed to collect RFP+ tumor cells after 48 h of mono- or co-culture conditions. **f** Sphere-formation assay of BxPC-3-RFP cells in mono-cultured and direct co-cultured conditions treated with IgG (2 μg/ml) or anti-activin A-neutralizing antibody (1 and 2 μg/ml). Values were presented as mean ± SD ($n = 4$). **g, h** 3D spheroid invasion assay of BxPC-3-RFP spheroids in mono-cultured and direct co-cultured conditions treated with IgG (4 μg/ml) or anti-activin A-neutralizing antibody (2 and 4 μg/ml). **g** Representative images. Scale bar, 200 μm. **h** Extensive rate of each tumor spheroid was calculated by normalizing RFP+ area at 72 h to RFP+ area at 1 h (left panel) and by normalizing RFP+ maximal diameter at 72 h to RFP+ maximal diameter at 1 h (right panel). Values were presented as mean ± SD ($n = 9$). **i, j** Subcutaneous injection of BxPC-3 cells and mixed fibroblasts/BxPC-3 cells treated with IgG (1 μg/ml) or anti-activin A-neutralizing antibody (1 μg/ml) every 3 days and collected tumor samples at day 14 after inoculation. Representative images **i** and quantification results of tumor weight **j**. Scale bar, 1 cm. Values were presented as mean ± SD ($n = 6$). **k, l** Subcutaneous injection of BxPC-3 cells and mixed fibroblasts/BxPC-3 cells treated with recombinant human follistatin 20 ng/ml every 3 days and collected tumor samples at day 14 after inoculation. Representative images **k** and quantification results of tumor weight **l**. Scale bar, 1 cm. Values were presented as mean ± SD ($n = 5$). **m, n** Quantitative real-time PCR analyses and western blotting analysis of ACTA2 and COL1A1 expression were performed in PSCs treated with recombinant human activin A (50 and 100 ng/ml) compared to untreated cells for 24 h and 48 h, respectively. For qPCR analyses, values were presented as mean ± SD ($n = 3$). **o, p** Quantitative real-time PCR analyses and western blotting analysis of ACTA2 and COL1A1 expression in mono-cultured, indirect co-cultured (ID), and direct co-cultured (D) fibroblasts (PSC-GFP) with tumor cells (BxPC-3). FACS was performed to collect GFP+ fibroblasts after 24 h and 48 h for mRNA and protein samples, respectively. For qPCR analyses, values were presented as mean ± SD ($n = 3$). **q** Western blotting analysis of proteins harvested from mono-cultured and direct co-cultured (D) fibroblasts (PSC-GFP) treated with IgG (2 μg/ml) or anti-activin A-neutralizing antibody (1 and 2 μg/ml). FACS was performed to collect GFP+ fibroblasts after 48 h of mono- or co-cultured conditions. **r, s** BxPC-3 cells and mixed PSCs/BxPC-3 cells treated with IgG (2 μg/ml) or anti-activin A-neutralizing antibody (2 μg/ml) every 3 days and collected tumor samples at day 14 after inoculation. Representative IHC images **r** and H Score **s** of αSMA expression. Scale bar, 0.5 mm. Values were presented as mean ± SD ($n = 6$). **t, u** BxPC-3 cells and mixed PSCs/BxPC-3 cells treated with follistatin (20 ng/ml) every 3 days and collected tumor samples at day 14 after inoculation. Representative IHC images **t** and H score **u** of αSMA expression. Scale bar, 0.5 mm. Values were presented as mean ± SD ($n = 5$). *$p < 0.05$, **$p < 0.01$, ***$p < 0.001$, ****$p < 0.0001$ (two-tailed t-test).

PANC-1 cells) were calcium-dependent axonal guidance and CREB signaling (Fig. 7f). Consistently, intracellular Ca²⁺ concentration was specifically up-regulated in fibroblasts during direct co-culturing with ATP1A1-overexpressing tumor cells (Fig. 7g; Supplementary Fig. 7a), while inhibition of ATP1A1 interaction through silencing ATP1A1 in tumor cells diminished the intracellular $Ca^{2+}$ concentration (Fig. 7h). In addition, treatment with a calcium channel inhibitor that abolished $Ca^{2+}$ oscillations significantly suppressed activin A secretion (Fig. 7i). Therefore, ATP1A1-mediated calcium signaling regulates activin A production. It was noted that low-dose ouabain at picomolar to nanomolar concentrations binds to the extracellular domain of ATP1A1 resulting in ATP1A1-IP3R (inositol 1,4,5-triphosphate receptor) interactions for generating calcium oscillations and NF-κB activation[31,32]. We then used low-dose ouabain to treat fibroblasts and tumor cells and found that NF-κB signaling and activin A production were specifically induced in fibroblasts, but not in tumor cells (Supplementary Fig. 7b, c). These results suggest that ATP1A1 in the fibroblast membrane may re-organize upon direct contact with tumor cells expressing high levels of ATP1A1, thereby inducing calcium oscillations, NF-κB activation, and activin A secretion.

It was noted that activin A production is regulated by activation of the NF-κB pathway in normal and cancer cells[25,33]. Activation of the NF-κB transcriptional response is measured by phosphorylation of p65 at S536, which enhances the transcriptional activity of p65[34]. Consistently, phosphorylated p65 at S536 (p-p65) was significantly increased in fibroblasts after direct contact with BxPC-3 cells through ATP1A1 interaction (Fig. 7j, k; Supplementary Fig. 7d). In addition, the NF-κB inhibitor BAY117082 suppressed activin A production (Fig. 7l). These results suggest that activation of the NF-κB pathway occurs specifically in fibroblasts upon direct contact with ATP1A1-overexpressing tumor cells. Taken together, binding to high

amounts of ATP1A1 of tumor cells enhances the amount of ATP1A1 at adhesion junctions of fibroblasts and triggers calcium oscillations, NF-κB activation, and activin A production.

## Discussion

Crosstalk between tumor cells and fibroblasts is a critical factor affecting tumor heterogeneity, tumor progression, and metastasis. The biological ramifications of direct cell–cell contact between fibroblasts and tumor cells are increasingly being recognized to be vitally important, as are the effects of paracrine signaling. Herein, we showed that homophilic ATP1A1 binding between ATP1A1-overexpressing tumor cells and fibroblasts induce EMT of tumor cells and activation of myofibroblasts through activin A secretion from fibroblasts (Fig. 8). Thus, juxta-tumoral fibroblasts serve as a stromal niche to support EMT of tumor cells at the tumor invasion front and in the circulation, leading to worsened patient outcomes.

The notion that myofibroblasts have tumor-restraining effects was suggested by the study that depletion of proliferating αSMA⁺ myofibroblasts in a αSMA-tk (thymidine kinase) transgenic mice with ganciclovir resulted in undifferentiated PDAC and decreased survival[35] of the mice, suggesting that αSMA⁺ fibroblasts may restrain tumor progression. However, juxta-tumoral myofibroblasts possess stemness properties[36] with low proliferative rate. In fact, they were preserved upon depletion of proliferating αSMA⁺ myofibroblasts by ganciclovir as shown in the Fig. 3a of the original paper (ref.[35]). Therefore, whether these juxta-tumoral αSMA⁺ myofibroblasts have tumor restraining roles remains unclear. On the contrary, we identified a subset of juxta-tumoral myofibroblasts which promotes tumor malignancy. This finding is consistent with the observation that the binding of fibroblasts enhances tumor invasiveness in lung cancer[37], peritoneal metastasis of ovarian cancer cells[17], and survival of CTCs in the

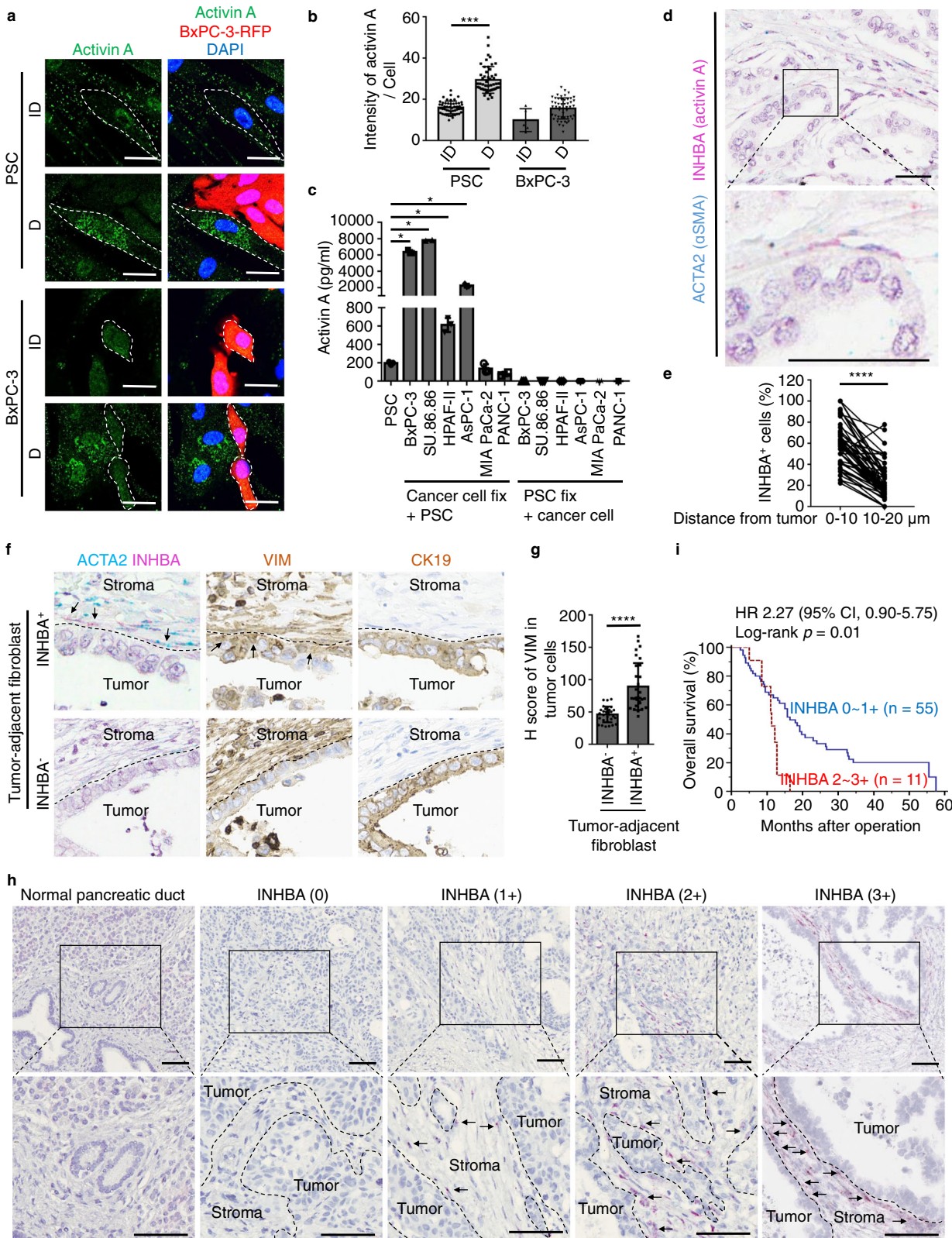

bloodstream[19]. The lack of consensus regarding tumor promoting or restraining roles of αSMA+ myofibroblast may be resolved when αSMA+ myofibroblasts can be further classified into several distinctive subtypes.

Cell-cell binding brings many membrane proteins in contact, including ligand-receptor pairs and junctional elements, which may exert additional effects in conjunction with adhesion and

signal transduction[38]. Contact-dependent communication transmits signals through direct pathway activation or juxtacrine signaling for short-range cell-to-cell communication, resulting in spatial tumor heterogeneity. In breast cancer, L1 cell adhesion molecule (L1CAM) expressed in cancer-associated fibroblasts stimulates ERK signaling via integrin α5β1 in cancer cells and promotes tumor growth[39]. In addition, fibroblasts activate

**Fig. 5 Activin A was mainly secreted from fibroblasts upon direct contact with tumor cells. a, b** Immunofluorescence staining of activin A in direct co-cultured PSCs and BxPC-3-cherry cells with or without contact with the other cell type measured by confocal microscope. A protein transport inhibitor (containing Monensin) was used for activin A accumulation in the Golgi complex before staining. Representative images **a** and quantification results of activin A intensity analyzed by Image J **b**. Scale bar, 5 μm. Values were presented as mean ± SD (n = 5 for BxPC-3 cells without direct contact with PSCs (due to low cell number); n = 50 for the other three groups). **c** ELISA analysis of activin A (R&D Systems) in supernatant of co-cultured fibroblasts with 2% paraformaldehyde fixed tumor cells (BxPC-3, SU.86.86, HPAF-II, AsPC-1, MIA PaCa-2, or PANC-1 cells), or reciprocally in supernatant of co-cultured tumor cells with 2% paraformaldehyde fixed fibroblasts. Values were presented as mean ± SD (n = 3). (**d**) Representative images of in situ hybridization (RNAscope, ACDBio) using probes for co-staining of activin A (INHBA, pink) and αSMA (ACTA2, blue) in PDAC patient specimen. Pink (activin A) and blue (αSMA) signal dots were co-localized in peri-tumoral cells. The nucleus was stained with hematoxylin. Scale bar, 50 μm. **e** Percentage of INHBA+ cells in regions with 0-10 μm and 10-20 μm distance from tumor–stromal interface in human PDAC specimens (n = 50). **f** Representative images of VIM and CK19 expression (brown color, IHC) and ACTA2/INHBA co-expression (ACTA2, αSMA, blue; INHBA, activin A, pink; RNAscope) in serial sections of human PDAC specimens. The nucleus was stained with hematoxylin. Scale bar, 50 μm. **g** H score of VIM expression levels in tumor cells with tumor-adjacent INHBA⁻ or INHBA⁺ CAFs. Values were presented as mean ± SD (n = 30). **h** Representative images of in situ hybridization (RNAscope) using a probe for the single stain of INHBA in PDAC patient specimen. The nucleus was stained with hematoxylin. Scale bar, 100 μm. The expression levels of INHBA in tumor-adjacent stromal cells were quantified as 0 (no expression), 1+ (less than 10% of tumor-adjacent cells), 2+ (10–50% of tumor-adjacent cells), and 3+ (>50% of tumor-adjacent cells). **i** Kaplan–Meier analysis of overall survival according to INHBA expression levels in regions within 10 μm distance from tumor (p = 0.01, Log-rank test). For statistical analyses, the score of 0 and 1+ were combined as the low expression group, and the score of 2+ and 3+ were combined as the high expression group. Blue line, INHBA^low in 0–10 μm, n = 55; Red line, INHBA^high in 0–10 μm, n = 11. *p < 0.05, ***p < 0.001, ****p < 0.0001 (two-tailed t-test).

NOTCH3 signaling to generate chemoresistant clones[40]. Furthermore, surface-associated FGF2 of fibroblasts binds to FGFR on colorectal tumor cells to induce tumor migration[41]. Similarly, we found that ATP1A1 serves simultaneously as a signaling receptor and adhesion molecule during tumor-fibroblast contact. This finding explains a common phenomenon that EMT occurs at the tumor invasion front along with the tumor-stromal interface and is sustained in circulating tumor cells[42].

Activin A as an autocrine activator of myofibroblast proliferation and differentiation[43] also modulates spatial heterogeneity of the tumor microenvironment. It is noted that activin A up-regulates ITGA11[44], which is required for myofibroblast activation[45]. Our observation that αSMA was highly expressed in peritumoral fibroblasts may be due to the autocrine activity of activin A. Activin A secreted at the tumor-stromal interface may also affect immune and endothelial cells by regulating differentiation of macrophage and dendritic cells, conversion of native CD4⁺ T cells to Foxp3⁺ Treg cells[46], and tube formation of endothelial cells[47]. Thus, it is plausible to suggest that secretion of activin A may alter the tumor microenvironment, especially along the tumor-stromal interface, to promote tumor progression.

In general, heterocellular interactions are less stable than homocellular interactions[48,49]. Nevertheless, there are sufficient cell–cell affinities to form heterocellular aggregates that can enter the circulation. As shown in Fig. 6b, multiple tumor membrane proteins that bind to fibroblasts were identified, indicating direct tumor-fibroblast contact is formed through membrane protein complexes that include adhesion molecules. Interestingly, it was noted that, in squamous cell carcinoma, E-cadherin and N-cadherin form a heterophilic linkage between tumor cells and CAFs that drives tumor invasion[50]. Similarly, breast cancer cells physically associate with fibroblasts through CDH11-CDH11 adhesion molecules to promote distant metastasis[51]. In the present study, we found that ATP1A1 from tumor cells binds fibroblast ATP1A1 at the plasma membrane for adhesion and induces downstream signaling. ATP1A1 is an α1 subunit of Na⁺/K⁺-ATPases, which consists of a catalytic α subunit and a regulatory β subunit to form the αβ enzyme complex[26,49]. Since ATP1A1 lacks a prominent extracellular domain, cell adhesion may depend on activation of ATP1A1 signaling to redistribute cell adhesion molecules[52] and increase expression of Na⁺/K⁺-ATPase β subunit[50,51], which has a highly glycosylated extracellular domain for cell-cell adhesion[53]. Thus, it is likely that the adhesion complex mediating binding between the tumor cells and

fibroblasts in PDAC consists of several other proteins, which warrants further investigation.

Intriguingly, homophilic ATP1A1 interaction turns on activin A secretion from fibroblasts, but not tumor cells. We also noted that ouabain induces NF-κB activation and activin A production only in fibroblasts but not in tumor cells (Supplementary Fig. 7b, c). This suggests that (1) ATP1A1 in fibroblasts is ready for binding to ouabain and (2) ouabain induces a conformational change in ATP1A1 that can turn on downstream signaling. On the contrary, ATP1A1 in tumor cells loses these properties. ATP1A1 of tumor cells may cooperate with other membrane proteins to bind to fibroblast's ATP1A1. These membrane protein(s) may likely mask tumor cells' ouabain binding site or alter ATP1A1's conformation, resulting in the loss of these activities in tumor cells. On the other hand, we observed that re-organization of fibroblast ATP1A1 with binding to tumor ATP1A1 complex induced calcium flux and activated the NF-κB pathway (Fig. 7). Further investigation to decipher the precise molecular mechanism of how homophilic ATP1A1 binding triggers NF-κB activation is warranted.

ATP1A1 is overexpressed in many cancer types, including hepatocellular carcinoma[54], non-small cell lung cancer[55], glioblastoma[56], breast cancer[57], and melanoma[58]. Knockdown of ATP1A1 decreased proliferation, migration, and tumorigenicity in hepatocellular carcinoma and squamous cell carcinoma[54,59]. Targeting ATP1A1 with doxorubicin-encapsulated nanoparticles with a peptide against ATP1A1 improved anti-tumor efficacy in breast cancer[60,61]. Our study further reveals the role of ATP1A1 in heterocellular adhesion and induction of juxtacrine activin A. Therefore, ATP1A1 could be a suitable target for moderately differentiated tumor cells such as BxPC-3 and SU.86.86 cells, which exhibit ATP1A1 overexpression with enhanced mesenchymal properties for metastasis.

CTMs, representing an intermediate status of metastasis, disseminate from tumor parenchyma and travel in the circulation as clusters (≥ 2 cells)[62]. CTMs form by persistent cell-cell adhesion, which greatly contributes to their survival from shear stress and anoikis[19]. It is hypothesized that CTMs containing stromal and immune cells aggregated with tumor cells may provide an additional advantage through paracrine interactions and immune evasion[62]. It was reported that treatment with Na⁺/K⁺-ATPase inhibitors leads to CTM cluster dissociation through down-regulation of stemness-related transcription factors, such as OCT4, SOX2, NANOG, and SIN3A[63], with subsequent suppression of lung metastasis[63]. As described in this report, the mechanism that facilitates the formation of circulating tumor-

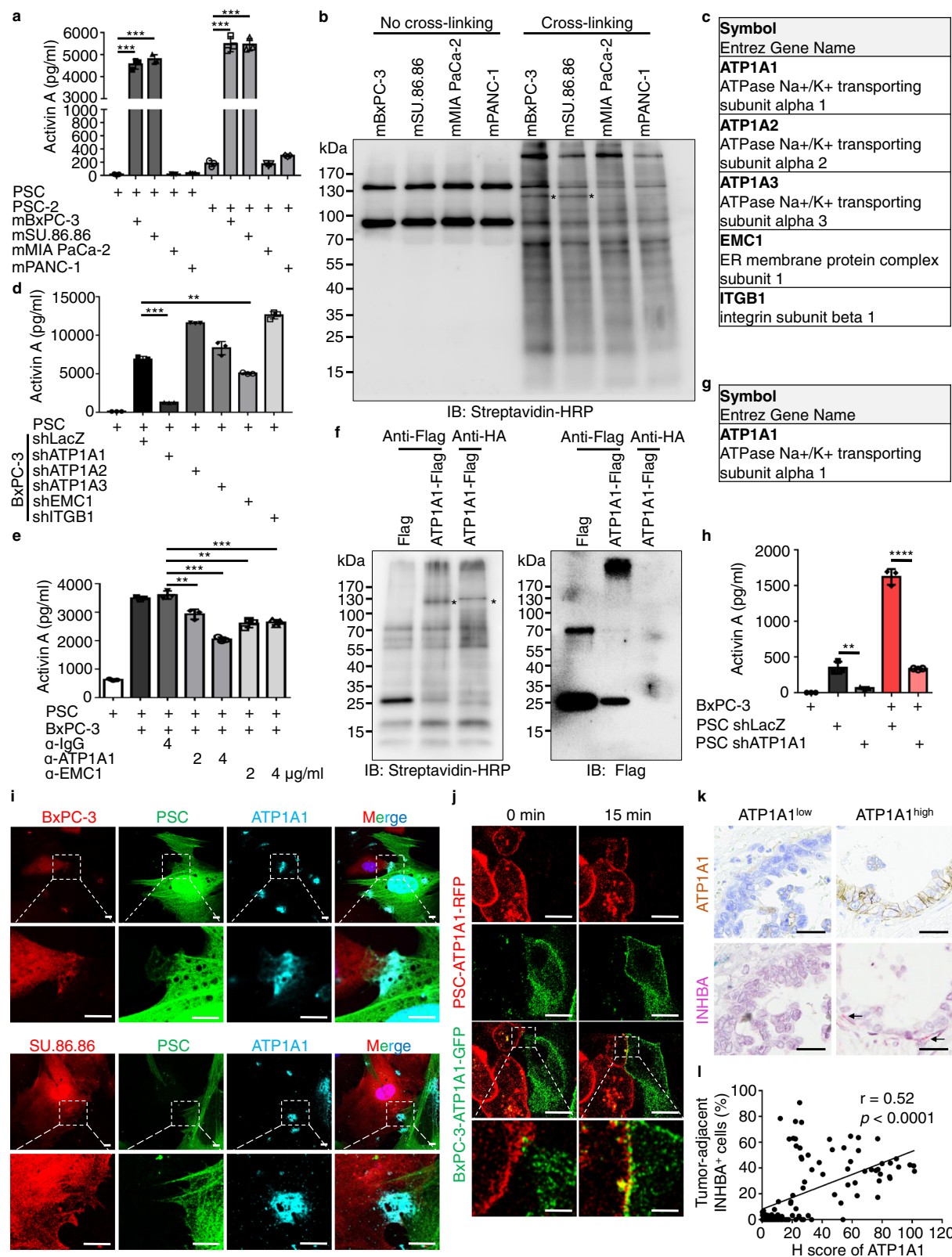

fibroblast clusters through homophilic ATP1A1 interactions is of importance in PDAC. This presents an intriguing opportunity to eliminate early metastasis by blocking this pathway.

Activin A is involved in diverse tumor-promoting functions[64,65], and targeting activin A may be a promising anticancer therapy. However, tumor and stromal cells (including fibroblasts, endothelial cells, and immune cells) are all capable of producing and responding to activin A forming a complex signaling network. In PDAC, activin A is secreted by both tumor cells and PSCs leading to overexpression in tumor ecosystems[64,66]. Our RNAscope and ELISA experiments clearly demonstrate that fibroblasts generate large amounts of activin A during direct tumor-fibroblast contact along with the tumor-stroma interface. Therefore, a combination of anti-activin A and anti-CAF therapy may improve outcome in a manner similar to

**Fig. 6 Homophilic ATP1A1 interactions promote activin A secretion. a** ELISA analysis of activin A in fibroblasts generated from two distinct individuals (PSC and PSC-2) after direct contact with purified tumor membrane proteins (from BxPC-3, SU.86.86, MIA PaCa-2, or PANC-1 cells). Values were presented as mean ± SD ($n = 3$). **b** Western blotting analysis of no cross-linking control and cross-linked membrane protein complexes with streptavidin-HRP under reducing condition. Since 2-mercaptoethanol broke disulfide bond on the cross-linker, a band (*) of biotin-labeled tumor membrane proteins from BxPC-3 and SU.86.86 cells were identified. **c** Five plasma membrane proteins were identified by mass spectrometry analysis. **d** ELISA analysis of activin A in mono-cultured or direct co-cultured PSCs with BxPC-3 cells stably expressing lentiviral-based LacZ[shRNA], ATP1A1[shRNA Clone#1], ATP1A2[shRNA], ATP1A3[shRNA], EMC1[shRNA Clone#1], ITGB1[shRNA]. Values were presented as mean ± SD ($n = 3$). **e** ELISA analysis of activin A in fibroblasts co-cultured with 2% paraformaldehyde fixed BxPC-3 cells and treated with IgG (4 μg/ml), anti-ATP1A1-neutralizing antibody (2 and 4 μg/ml, 14418-1-AP, Proteintech), or anti-EMC1-neutralizing antibody (2 and 4 μg/ml, GTX119884, Genetex). Values were presented as mean ± SD ($n = 3$). **f** Left panel, western blotting analysis of biotinylated fibroblast membrane proteins with streptavidin-HRP under reducing condition. A band (*) of biotin-labeled fibroblast membrane proteins which bound to ATP1A1 was identified. Right panel, re-probing the western blot membrane from left panel with anti-Flag antibody to confirm that the identified band (*) was not ATP1A1-Flag. **g** ATP1A1 was identified by mass spectrometry analysis. **h** ELISA analysis of activin A in mono-cultured and direct co-cultured BxPC-3 cells with PSCs stably expressing lentiviral-based LacZ[shRNA] and ATP1A1[shRNA Clone#1]. Values were presented as mean ± SD ($n = 3$). **i** Immunofluorescence staining of ATP1A1 in fibroblasts (PSC-GFP) and cancer cells (BxPC-3-RFP and SU.86.86-RFP) measured by confocal microscope. Scale bar, 2 μm. **j** Live-cell confocal imaging of BxPC-3 cells stably expressing ATP1A1-GFP and PSCs stably expressing ATP1A1-RFP. Co-localization of ATP1A1 was compared between initiation and establishment of tumor-fibroblast contact within 15 mins. Scale bar, 5 μm. **k, l** Correlation of ATP1A1 and INHBA expression levels in PDAC patient specimen. **k** Representative images of ATP1A1 expression (brown color, IHC) and INHBA expression (activin A, pink, in situ hybridization, RNAscope). Scale bar, 25 μm. **l** Pearson's correlation coefficient analysis of ATP1A1 and INHBA according to H score and tumor-adjacent INHBA+ cells (%) ($n = 100$, $r = 0.52$, $p < 0.0001$). **$p < 0.01$, ***$p < 0.001$, ****$p < 0.0001$ (two-tailed t-test).

how the addition of a smoothened inhibitor against the hedgehog pathway significantly improves the tumor suppression effect of SB431542, which targets the TGF-β/Nodal/Activin pathways[67].

Our study highlights the importance of tumor-supporting niches modified by direct contact between ATP1A1-overexpressing tumor cells and fibroblasts. Homophilic ATP1A1 interaction facilitates heterotypic cell adhesion and directly regulates cytoplasmic signal transduction. This tumor-promoting cell-cell interaction induces juxtacrine activin A to generate tumor cells with EMT plasticity and autocrine activin A for fibroblast self-activation. Tumor-fibroblast clusters established at the tumor invasion front survive to reach the general circulation, thereby facilitating collective invasion, anoikis resistance, and metastasis of cancer cells (Fig. 8). Blockage of direct contact between tumor cells and fibroblasts may represent a promising therapeutic strategy for patients with PDAC.

## Methods

**Ethics statement and patient information in the cohort study.** All pancreatic cancer patient data and tissue specimens were acquired from the National Taiwan University Hospital (NTUH), Taipei, Taiwan and approved by the Institutional Review Board of the NTUH (201303029RINC, 201411085RINB, and 201701015RINA). The informed consent and permissions were obtained from human participants. Between 2007 and 2015, pancreatic cancer specimens were collected from 85 patients who underwent pancreaticoduodenectomy in NTUH. Based on the 7th edition of the American Joint Committee on Cancer (AJCC) criteria, PDAC patients were staged as IIA ($n = 32$), IIB ($n = 52$), and 1 unknown status (Table 1). For immunohistochemistry (IHC) analysis, total patient specimens were used. For in situ mRNA hybridization, patient specimens ($n = 19$) without positive control signals were removed in consideration of poor RNA quality.

**Cell lines.** All pancreatic cancer cell lines were purchased from ATCC. Human pancreatic cancer cell lines BxPC-3, SU.86.86, AsPC-1 were cultured in RPMI1640 medium. Human pancreatic cancer cell line HPAF-II was cultured in MEM medium. Human pancreatic cancer cell line MIA PaCa-2 and PANC-1 were cultured in DMEM medium. All the culture medium was supplemented with 10% fetal bovine serum (FBS), 2 mM L-glutamine, 1 mM non-essential amino acid (NEAA), 1 mM sodium pyruvate, penicillin and streptomycin (100 IU/ml and 100 μg/ml, respectively). The culture medium of MIA PaCa-2 added an extra 5% horse serum. Mouse primary pancreatic stellate cells were isolated from 8 weeks old C57BL/6 mice using Nycodenz gradient method[68]. Human primary pancreatic stellate cells (PSC) were obtained from ScienCell Research Laboratories, inc., and patient-derived primary pancreatic fibroblasts (PSC-2) were isolated from a human specimen with IPMN lesion and without PDAC. All fibroblasts were cultured in stellate cells medium (ScienCell Research Laboratories, inc.). For co-culture experiments, both tumor cells and fibroblasts were cultured in DMEM/F12 medium supplemented with 10% FBS, 2 mM L-glutamine, 1 mM NEAA, 1 mM sodium pyruvate, penicillin, and streptomycin (100 IU/ml and 100 μg/ml, respectively).

**Immunofluorescence staining in formalin-fixed paraffin-embedded (FFPE) tissue sections.** For immunofluorescence (IF) co-staining, the formalin-fixed paraffin-embedded (FFPE) whole tissue sections (4 μm thick) were processed for IF studies. Staining was performed on an automated staining system (Ventana Benchmark LT, Ventana Medical Systems or Bond-max, SP8systems) followed by manually added antibodies and DAPI. For co-staining of αSMA and CK19 in human specimens and mouse pancreases, anti-αSMA antibody (1:200; ab7817, Abcam) and anti-CK19 antibody (1:200; GTX112666, Genetex) were used. For co-staining of VIM and CK19 in human specimens and mouse pancreases, anti-vimentin antibody (1:500; 10366-1-AP, Proteintech) and anti-CK19 antibody (1:150; GTX27755, Genetex) were used. For co-staining of VIM and αSMA in human specimens and mouse pancreases, anti-vimentin antibody (1:500; 10366-1-AP, Proteintech) and anti-αSMA antibody (1:1500; ab7817, Abcam) were used. Slides were incubated with primary antibodies for 4 °C overnight, and then incubated with fluorescent secondary antibodies (1:200; Alexa Fluor 488/594/647, Invitrogen) for 1 h at room temperature. For co-staining of CLDN1/CK19/αSMA or VIM/CK19/αSMA in xenograft tumors of BxPC-3 cells co-injected with PSCs, anti-CLDN1 antibody (1:100; 51-9000, Invitrogen) or anti-VIM antibody (1:500; 10366-1-AP, Proteintech), and anti-CK19 antibody (1:150; GTX27755, Genetex) were incubated for 4 °C overnight. After fluorescent secondary antibodies (1:200; Alexa Fluor 594/647, Invitrogen) incubation, specimens were finally incubated with anti-αSMA-Alexa Fluor 488-conjugated antibody (1:100; 53-9760-82, Invitrogen).

**IHC staining and image assessment.** The formalin-fixed, paraffin-embedded whole tissue sections (4 μm thick) were processed for IHC studies. IHC for each antigen was performed on an automated staining system (Ventana Benchmark LT, Ventana Medical Systems or Bond-max, SP8systems) following manufacturer's instructions. The sections were counter-stained with hematoxylin. For staining of αSMA, anti-αSMA antibodies (1:400; ab7817, Abcam) were used for human specimens, and anti-αSMA antibodies (1:400; NB600-531, Novus biological) were used for mouse pancreas. For staining of vimentin, anti-vimentin (1:2000; 10366-1-AP, Proteintech) antibodies were used for human specimens and mouse pancreases. For staining of ATP1A1, anti-ATP1A1 (1:200; GTX635461, Genetex) antibodies were used for human specimens. For the negative controls, the primary antibodies were replaced with normal mouse IgG and normal rabbit IgG. Expression levels of αSMA, vimentin, and ATP1A1 were scored by Aperio ImageScope (Aperio, USA) using the positive pixel count v9 algorithm. For each section, at least 10 randomly selected 400× high-power fields were used, and stromal regions of 0–10 μm and 10–20 μm distance from tumor margins were circled as annotated layers for analysis. H-score was assigned by [1 × (% 1+ cells) + 2 × (% of 2+ cells) + 3 × (% of 3+ cells)].

The association between ATP1A1 and activin A (INHBA) expression was measured in 5 PDAC patient specimens collected at NTUH. For each case, 20 randomly selected 400× high-power field regions with their corresponding regions in serial sections were stained with ATP1A1 (by IHC) and INHBA (by RNAscope). To measure the expression level of ATP1A1, each 400× high-power field cancer-cell only regions were outlined with pen tools and analyzed by the positive pixel count v9 algorithm of Aperio ImageScope software (Aperio, USA) to calculate the average percentage of 1+, 2+, and 3+ cells. H-score was assigned by [1 × (% of 1+ cells) + 2 × (% of 2+ cells) + 3 × (% of 3+ cells)]. For the corresponding expression levels of INHBA, number of INHBA positive and negative nucleus in stromal regions of 0 ~ 10 μm distance from tumor margins were measured, and the percentage of INHBA positive nucleus was calculated.

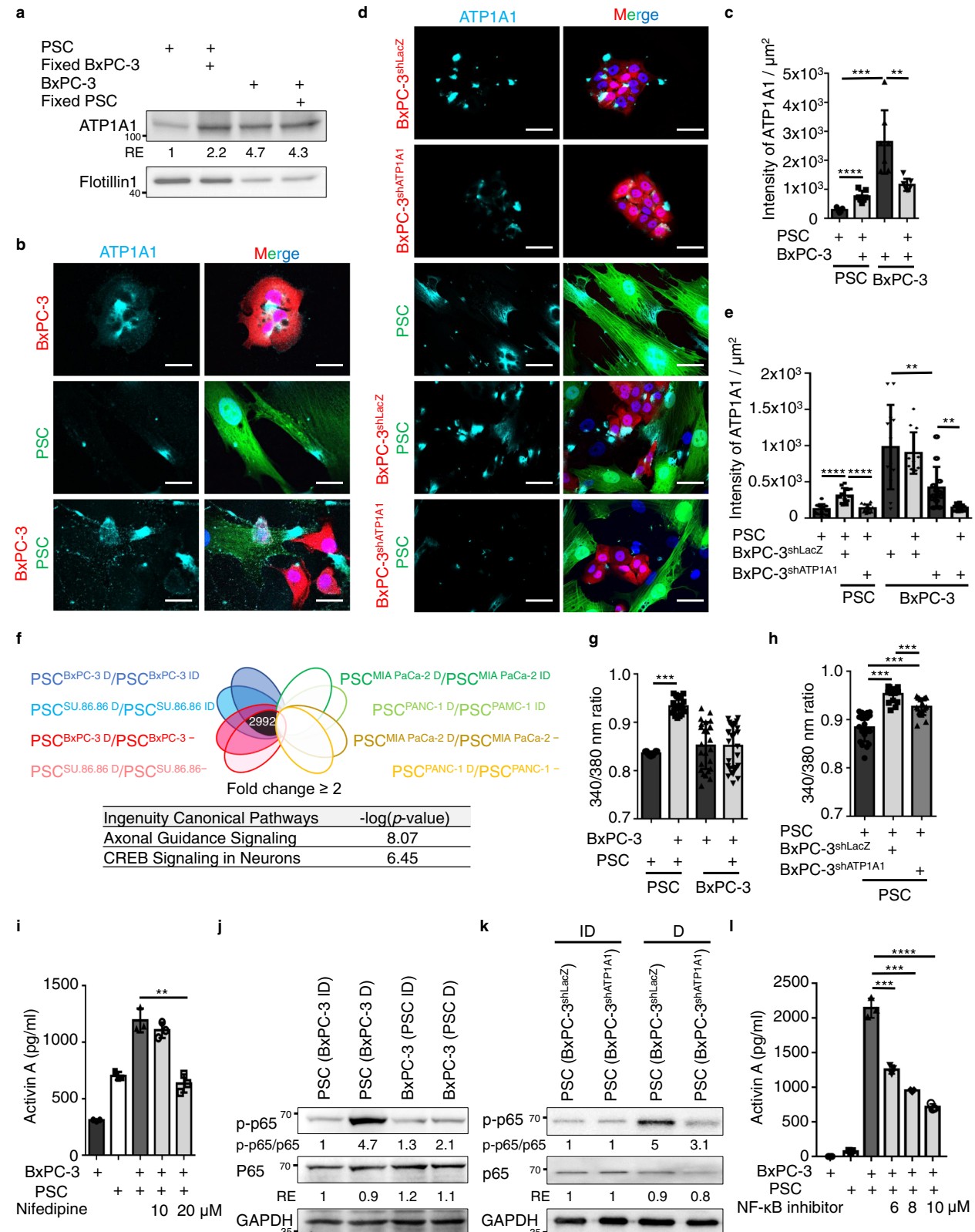

## MACS dissociation, fluorescence-activated cell sorting (FACS), and IF staining

Pancreases were isolated from 6 to 7 weeks old *Pdx1-Cre*; LSL-*Kras*[+/+]; *Trp53*[flox/flox] (WT) and *Pdx1-Cre*; LSL-*Kras*[G12D/+]; *Trp53*[flox/flox] (PdKP53) transgenic mice. The total pancreas was cut into pieces of less than 1 mm³ and underwent enzymatic digestion using collagenase P (1 mg/ml; Roche) and soybean trypsin inhibitor (0.1 mg/ml; Gibco) in HBSS (with Ca²⁺ and Mg²⁺) at 37 °C for 25 min with intermittent pipetting. After neutralization with DMEM/F12 medium (supplemented with 10% fetal calf serum and 10 µg/ml antibiotics/antimycotics),

tissue fragments with the medium were transferred to a mechanical dissociator C tube. The mechanical dissociator program A.01 was used twice (one cycle = 25 s). The homogenate was passed through a 40 µm cell strainer and centrifuged at 300 × *g* at room temperature for 5 mins. After washing the pellet with PBS, the dissociated cells were stained with anti-EpCAM antibody (1:200; 14-5791-85, ebioscience) and anti-PDGFRα antibody (1:200; ab48202, Abcam) for 1 h at 4 °C. After washing, cells were incubated with secondary antibodies (1:200; anti-rat Alexa Fluor 488 and anti-mouse Alexa Fluor 647) and eFluor 780 viability dye

**Fig. 7 Homophilic ATP1A1 interactions stabilize ATP1A1 in plasma membrane of fibroblasts trigger intracellular Ca$^{2+}$ oscillations for turning on activin A secretion via NF-κB signaling. a** Western blotting analysis of plasma membrane proteins from intracellular hydrogelated PSCs and BxPC-3 cells. Before intracellular hydrogelation, cells were cultured with or without 2% paraformaldehyde fixed BxPC-3 cells or PSCs for 24 h. **b** Immunofluorescence staining of ATP1A1 in fibroblasts (PSC-GFP) and cancer cells (BxPC-3-RFP) upon mono-cultured or direct co-cultured conditions measured by confocal microscope. Scale bar, 10 μm. **c** Quantification results of ATP1A1 intensity analyzed by Image J. Values were presented as mean ± SD (n = 7). **d** Immunofluorescence staining of ATP1A1 in fibroblasts (PSC-GFP) and cancer cells (BxPC-3-RFP cells stably expressing lentiviral-based LacZ$^{shRNA}$ or ATP1A1$^{shRNA Clone#1}$) upon mono-cultured or direct co-cultured conditions measured by confocal microscope. Scale bar, 10 μm. **e** Quantification results of ATP1A1 intensity analyzed by Image J. Values were presented as mean ± SD (n = 12). **f** The differential gene expression analysis with RNA-seq. Overlap of the commonly up-regulated genes (2992) in fibroblasts (PSC-GFP) after direct co-cultured (D) with BxPC-3 and SU.86.86 cells (but not MIA PaCa-2 and PANC-1 cells) compared to mono-cultured (−) or indirect co-cultured (ID) conditions were used for Ingenuity Pathway Analysis (IPA). Top 2 canonical pathways with the largest -log (p-value) calculated by Fisher's Exact Test in IPA were shown. **g** Intracellular Ca$^{2+}$ concentration of BxPC-3 cells and PSCs in mono-cultured or direct co-cultured conditions measured by fluorescence ratios (340/380 nm) using Fura-2/AM. Values were presented as mean ± SD (n = 20). **h** Intracellular Ca$^{2+}$ concentration of PSCs cells upon mono-cultured or direct co-cultured with BxPC-3 cells stably expressing lentiviral-based LacZ$^{shRNA}$ or ATP1A1$^{shRNA clone#1}$ measured by fluorescence ratios (340/380 nm) using Fura-2/AM. Values were presented as mean ± SD (n = 20). **i** ELISA analysis of activin A in mono-cultured or direct co-cultured PSCs and BxPC-3 cells with Ca channel inhibitor (Nifedipine). Values were presented as mean ± SD (n = 3). **j** Western blotting analysis of proteins harvested from co-cultured fibroblast-GFP with BxPC-3 cells. FACS was performed to collect GFP$^+$ and GFP$^-$ cells after 48 hours of indirect or direct co-culture. **k** Western blotting analysis of proteins harvested from indirect- and direct co-cultured PSC cells with BxPC-3-RFP stably expressing lentiviral-based LacZ$^{shRNA}$ and ATP1A1$^{shRNA Clone#1}$. FACS was performed to collect RFP$^-$ cells after 48 hours of indirect (ID)- or direct (D)- co-cultured conditions. **l** ELISA analysis of activin A in mono-cultured or direct co-cultured PSCs and BxPC-3 cells with NF-κB inhibitor (BAY117082). Values were presented as mean ± SD (n = 3). **p < 0.01, ***p < 0.001, ****p < 0.0001 (two-tailed t-test).

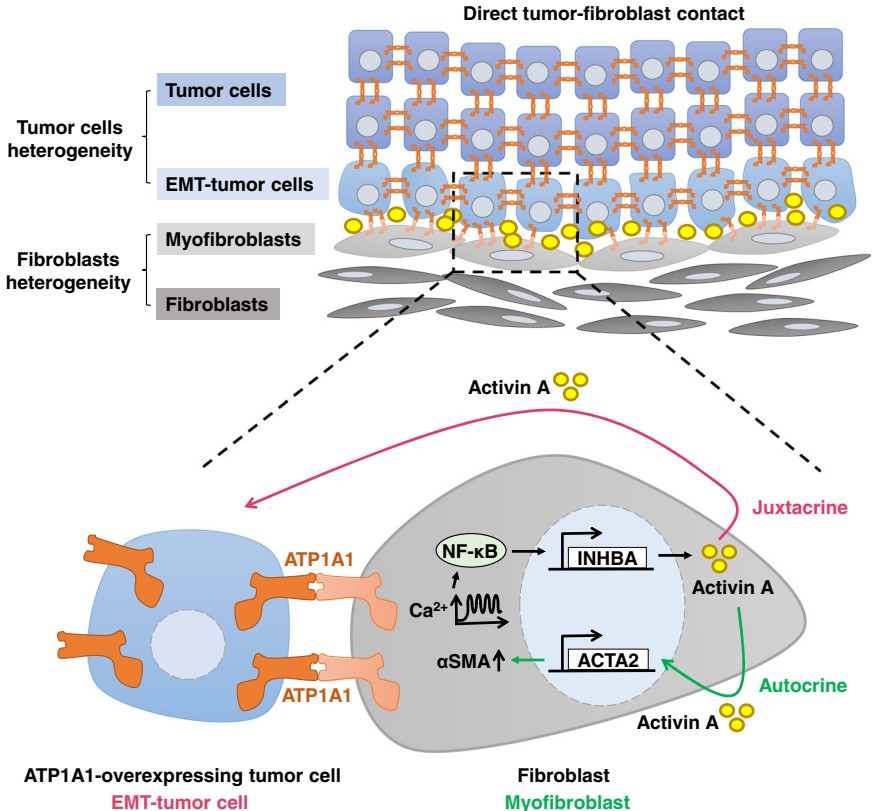

**Fig. 8 Proposed model of homophilic ATP1A1 interactions trigger activin A secretion to promote EMT of tumor cells and myofibroblast activation for tumor invasion and metastasis.** Homophilic ATP1A1 interactions between ATP1A1-overexpressing tumor cells and fibroblasts stabilize ATP1A1 on fibroblasts leading to calcium oscillations, NF-κB activation, and activin A secretion. High concentration of activin A along tumor-stromal interface induces EMT of tumor cells for tumor invasion and metastasis. On the other hand, autocrine activin A activates a subpopulation of peri-tumoral αSMA$^+$ myofibroblast.

(1:1000; 65-0865-14, ebioscience) for FACS. Cell clusters were gated by the width (W) parameter.

For immunofluorescence (IF) co-staining of αSMA and CK19 in sorted tumor-fibroblast clusters, cell clusters were plated on poly-L-lysine-coated glass slides and fixed with 4% paraformaldehyde. After permeabilization with 0.5% TritonX-100 and blocking with 1% goat serum, cells were subjected to IF staining with anti-CK19 antibody (1:100; GTX112666, Genetex) at 4 °C overnight. After washing with PBST (PBS + 0.1% Tween20), cells were incubated with secondary antibodies (1:200; anti-rabbit Alexa Fluor 594) for 1 h at room temperature. After washing

with PBST, cells were incubated with anti-αSMA-Alexa Fluor 488 antibody (1:100; 53-9760-82, Invitrogen) for 1 h at room temperature. The nucleus was stained with DAPI.

**Isolation and imaging of circulating tumor microemboli (CTM).** Peripheral blood from 6 to 7 weeks old PdKP53 transgenic mice and PDAC patients were collected in Na-Heparin tubes. The blood of 3 PdKP53 mice was pooled for one sample to obtain enough blood for Accuspin. Blood was poured into Accuspin

tubes (Sigma) to perform density gradient separation by Histopaque-1077. After centrifugation, circulating tumor cells together with mononuclear cells were isolated at the plasma-Histopaque-1077 interface. After three times of PBS washes, cells were fixed with 4% paraformaldehyde for 10 mins. CD45$^+$ leucocytes were depleted by Dynabeads CD45 (Invitrogen) and placed on slides by Cytospin (Shandon Cytospin 4 Centrifuge, Thermo Scientific) at 500 rpm for 5 min. Cells on the Cytospin slide were fixed with 4% paraformaldehyde for 20 mins. After a gentle rinse with PBS, cells were permeabilized with 0.2% TritonX-100 for 10 mins at room temperature.

For IF co-staining of αSMA, CK19, and CD45 in CTM from 6 to 7 weeks old PdKP53 mice, cells were blocked with 1% FBS for 1 h at room temperature and Fab fragment (200 μg/ml, Jackson ImmunoResearch Laboratories) for 2 h at room temperature. After gentle rinse with PBST (PBS + 0.1% Tween20), cells were incubated with anti-CD45 antibody (1:100; 14-0452-85, eBioscience) and anti-CK19 antibody (1:100; GTX112666, Genetex) simultaneously for 4 °C overnight. After a gentle rinse with PBST, cells were incubated with secondary antibody (1:200; anti-rabbit Alexa Fluor 594 and anti-rat Alexa Fluor 647) for 1 h at room temperature. After a gentle rinse with PBST, cells were incubated with anti-αSMA-Alexa Fluor 488 antibody (1:100; 53-9760-82, Invitrogen) for 1 h at room temperature. The nucleus was stained with DAPI.

For IF co-staining of αSMA, CK19, and CD45 in CTC clusters from PDAC patients, cells were blocked with 1% FBS for 1 h at room temperature. Cells were incubated with anti-CD45 antibody (1:50; 14-0452-85, eBioscience), anti-CK19 antibody (1:50; GTX112666, Genetex), and anti-αSMA antibody (1:50; ab7817, Abcam) simultaneously for 2 h at room temperature. After a gentle rinse with PBST, cells were incubated with secondary antibody (1:100; anti-rat Alexa Fluor 647, anti-rabbit Alexa Fluor 594, and anti-mouse Alexa Fluor 488) for 1 h at room temperature. After a gentle rinse with PBST, cells were stained with DAPI. Fluorescent signals of whole Cytospin regions were captured by Leica SP8 confocal via Navigator module (LAS X).

**3D-Matrigel co-culture assay.** Human pancreatic tumor cells labeled with RFP and mouse pancreatic tumor cells labeled with GFP were co-cultured with human pancreatic stellate cells (PSC-GFP) and mouse pancreatic stellate cells (mPSC-RFP), respectively. Mixed tumor cells and fibroblasts were seeded onto the polymerized Matrigel (4.5 mg/ml) and covered with DMEM/F12 medium (supplemented with 10% fetal bovine serum and 10 μg/ml antibiotics/antimycotics) containing 0.2 mg/ml Matrigel. After 2 and 24 h, the cells were subjected to Hoechst staining and imaged by inverted fluorescence microscope (Olympus). The percentage of cancer cells and fibroblasts with distance ≤ 1 cell diameter was determined by Imaris software (Oxford instruments) at indicated time points.

**Sphere-formation assay.** In a 6-well ultra-low attached dish, 100 tumor cells (BxPC-3-RFP, SU.86.86-RFP, PdKP53-GFP, or KPC-GFP) and 100 or 300 fibroblasts (PSC-GFP or mPSC-RFP) were cultured in mono-cultured, indirect (transwell) co-cultured, or direct co-cultured manners in DMEM/F12 medium (supplemented with 10% fetal calf serum and 10 μg/ml antibiotics/antimycotics). For indirect (transwell) co-culture, fibroblasts were seeded in 0.4 um cell culture insert (Corning), and tumor cells were seeded in ultra-low attached dish below the insert. To enhance cell-cell adhesion, all the experimental cells in the direct co-cultured group (tumor cells, fibroblasts, or mixed tumor cells and fibroblasts) were pelleted down and kept in room temperature for 30 min before seeding. Cells were maintained in a humidified 37 °C incubator for 7 days. The number of RFP$^+$ spheroids (human cell lines) and GFP$^+$ spheroids (mouse cell lines) were calculated with a fluorescence microscope.

**3D spheroid invasion assay.** A total of $10^4$ BxPC-3-RFP cells were seeded in ultra-low attached round-bottomed 96-well plates (Corning) for 2 days. The experiment was performed on 96 well flat-bottom plates (Corning). Matrigel (4.5 mg/ml) was generated by 1:1 mixture of Matrigel (9 mg/ml) and DMEM/F12 medium (supplemented with 10% fetal bovine serum and 10 μg/ml antibiotics/antimycotics). A schematic illustration of the experimental procedure was shown in Fig. 3c. For mono-cultured conditions, tumor spheroids were seeded on the polymerized Matrigel (90 μl) and embedded in Matrigel (30 μl). For indirect co-culture conditions, $10^4$ fibroblasts were seeded on the polymerized Matrigel (30 μl) and embedded in Matrigel (30 μl). Another layer of Matrigel (30 μl) was seeded on top of the fibroblasts followed by tumor spheroids embedded in Matrigel (30 μl). For direct co-culture conditions, mixed tumor cells and fibroblasts were seeded onto the polymerized Matrigel (90 μl) and embedded in Matrigel (30 μl). To enhance cell-cell adhesion, mixed tumor cells and fibroblasts were pelleted down and kept in room temperature for 30 min before seeding. Microscopical images were captured after 1, 24, 48, and 72 h. The extensive rate was calculated by normalizing the area and maximal diameter of RFP$^+$ spheroid from 72 h to 1 h.

**Immunofluorescence staining of adherent cell cultures.** For IF staining of activin A, fibroblasts and BxPC-3-RFP cells were direct co-culture for 2 days. Six hours before fixation, BD GolgiStop$^{TM}$ Protein Transport Inhibitor (containing Monensin) (BD 554724) was added to culture medium. Then, cells were washed with PBS, fixed with 4% paraformaldehyde, and permeabilized with 0.5% TritonX-

100. After blocking with 5% FBS, cells were incubated with anti-activin A antibody (1:20; 10651-1-AP, proteintech) for 4 °C overnight. After wash with PBST (PBS + 0.1% Tween20), cells were incubated with secondary antibody (1:100; anti-rabbit Alexa Fluor 488) and nucleus was stained with DAPI.

For IF staining of ATP1A1, tumor cells (BxPC-3-RFP, BxPC-3-RFP$^{shLacZ}$, BxPC-3-RFP$^{shATP1A1}$, and SU.86.86-RFP) and PSC-GFP cells were mono-cultured or direct co-cultured for 2 days. Cells were washed with PBS and fixed with 4% paraformaldehyde. After blocking with 1% FBS, cells were incubated with anti-ATP1A1 antibody (1:50; 14418-1-AP, proteintech) for 4 °C overnight. After washing with PBS, cells were incubated with secondary antibody (1:200; anti-rabbit Alexa Fluor 647) for 1 h at room temperature and nucleus was stained with Hoechest 33342.

**Collagen gel contraction assay.** Collagen (Corning), 2X DMEM/F12 (Gibco), NaOH, and HBSS (with Ca$^{2+}$ and Mg$^{2+}$) were mixed together to a final concentration of 2 mg/ml of collagen. A total of $2 \times 10^4$ PSCs/250 μl collagen mixture were seeded in 24 well plate. The plate was incubated 1 h at 37 °C for collagen polymerization. BxPC-3 cells were seeded in 0.4μm cell culture insert (Corning) to separate from collagen gel for indirect co-culture. BxPC-3 cells were seeded on the collagen gel for direct co-culture. Each well was gently loaded with DMEM/F12 medium (supplemented with 2% fetal bovine serum and 10 μg/ml antibiotics/antimycotics), and the gels were dissociated from their mold by gently running the tip of a 200-μL pipet tip along gel edges.

**Isolation of membrane protein.** Membrane proteins were collected under sterile conditions. Tumor cells were incubated in hypotonic buffer (20 mM HEPES, 10 mM KCl, 1.5 mM MgCl$_2$, 1 mM EGTA, and protease inhibitor cocktail; pH 7.4) with dounce homogenization for osmotic lysis of plasma membrane. After centrifuging at $700 \times g$ at 4 °C for 5 min to remove nucleus and cell debris, the supernatant was mixed with 2.5 M sucrose to final 250 mM sucrose. Cytosolic and membrane fractions were ultracentrifuged at $100,000 \times g$ (28700 rpm) at 4 °C for 1 h. The membrane pellet was re-suspended in HEPES lysis buffer-1 (50 mM HEPES, 150 mM NaCl, 1% NP-40, 0.5% sodium deoxycholate, 0.1% SDS, and protease inhibitor cocktail; pH 7.4) and sonicated.

**Cross-linking of membrane proteins by Sulfo-SBED cross-linkers.** Every step was kept on ice and avoided light. Tumor membrane proteins (10 mg/ml, total 1500 μg) were mixed with Sulfo-SBED (Thermo), and incubated for 30 mins at room temperature. Desalting column (Zeba 7 K MWCO) was used to remove non-reactive cross-linkers. Tumor membrane proteins with cross-linkers were incubated with intracellular hydrogelated fibroblasts for 5 h at 4 °C. After HBSS (pH7.4, with Ca$^{2+}$ and Mg$^{2+}$) wash, cells were covered with HBSS (pH7.4, with Ca$^{2+}$ and Mg$^{2+}$) and photoactivated with 365 nm UV lamp (UVP CL1000 cross-linker) using 2800 μJ/cm$^2$ at a distance of 5 cm for 15 mins. Cells were lysed with HEPES lysis buffer-1 (50 mM HEPES, 150 mM NaCl, 1% NP-40, 0.5% sodium deoxycholate, 0.1% SDS, and protease inhibitor cocktail; pH 7.4) and mildly sonicated (for 1 min at 4 °C) to remove plasma membrane from gelated cytosol and nucleus. After centrifuging $700 \times g$ for 5 min, supernatant containing plasma membrane was collected and sonicated. Samples were pre-cleaned with protein A/G agarose beads (Thermo Scientific) for 2 h at 4 °C, and then incubated with monomeric-avidin beads (Thermo Scientifics) at 4 °C overnight. Biotinylated proteins were eluted with D-biotin (12.5 mM) for 2 h at 4 °C. Sample buffer was added for 30 mins at 40 °C before running SDS-PAGE.

**Intracellular hydrogelation.** The protocol was modified from the original article published by Lin et al.[29,30]. Tumor cells or fibroblasts in 10 cm dish were seeded 1 day before experiment. Gelation buffer was freshly prepared by mixing 1.8 ml poly (ethylene glycol) diacrylate (PEG-DA Mn = 700, Sigma-Aldrich), 45 mg 2-hydroxy-4′-(2-hydroxyethoxy)-2-methylpropiophenone (Irgacure D-2959, Sigma-Aldrich), 30 μl DMSO, and 1.2 ml HBSS (pH7.4, with Ca$^{2+}$ and Mg$^{2+}$). Fibroblasts were washed by PBS and incubated with gelation buffer for 5 mins at room temperature. Gelation buffer was removed from fibroblasts. Fibroblasts were immersed in HBSS (pH7.4, with Ca$^{2+}$ and Mg$^{2+}$) for UV cross-linking with 365 nm UV lamp (UVP CL1000 cross-linker) using 2800 μJ/cm$^2$ at a distance of 5 cm for 15 mins, and then were ready for following experiments.

**Cross-linking of ATP1A1 binding protein.** To discover the binding partner of ATP1A1 in fibroblasts, membrane protein fraction of HEK293T cells transfected with pCMV3-Flag (Sino Biological) or pCMV3-ATP1A1-Flag (Sino Biological) were purified by ultracentrifugation. Briefly, the membrane pellet was re-suspended in HEPES lysis buffer-2 (50 mM HEPES, 150 mM NaCl, 3% TritonX-100, 5% Glycerol, and protease inhibitor cocktail; pH 7.4), sonicated, pre-cleaned with protein A/G agarose beads (Thermo Scientific), and then incubated with M2 (Samples: Flag and ATP1A1-Flag; Sigma) or HA (Sample: ATP1A1-Flag; Sigma) beads at 4 °C overnight. ATP1A1-Flag membrane protein complexes were kept on agarose beads due to their low solubility.

PSCs were hydrogelated intracellularly, lysed with HEPES lysis buffer-2 (50 mM HEPES, 150 mM NaCl, 3% TritonX-100, 5% Glycerol, and protease inhibitor cocktail; pH 7.4), and mildly sonicated (for 1 min at 4 °C) to remove plasma

membrane from gelated cytosol and nucleus. After centrifuging at $700 \times g$ for 5 min, supernatant was collected and sonicated. Plasma membrane fraction of PSCs were incubated with Sulfo-NHS-biotin crosslinker (Thermo Scientific) at 4 °C overnight, and quench the reaction with 20 mM Tris-HCl (pH 7.5) at 4 °C for 15 min. ATP1A1-Flag-bound M2 agarose beads or control agarose beads (Flag-bound M2 beads and ATP1A1-Flag-bound HA beads) were incubated with biotin labeled plasma membrane proteins from intracellular hydrogelated fibroblasts in HEPES lysis buffer-3 (50 mM HEPES, 150 mM NaCl, 1% TritonX-100, 5% Glycerol, and protease inhibitor cocktail; pH 7.4) at 4 °C overnight. Potential interacting proteins of ATP1A1 were pull-down by agarose beads, and released by detergent (add 0.2% SDS and 1% Sodium deoxycholate) for further capturing by monomeric-avidin beads (Thermo Scientifics) and eluted by D-biotin (12.5 mM) within sample buffer at 40ºC for 30 mins twice.

**Calcium imaging.** Calcium imaging was performed according to the previous study[69]. To monitor the intracellular calcium concentration, mono-culture and direct co-culture of PSCs and BxPC-3-RFP cells were seeded on 15 mm cover slide in DMEM/F12 medium (supplemented with 2% fetal bovine serum and 10 μg/ml antibiotics/antimycotics) for 48 h. After HBSS wash, cells were loaded with 5 μM Fura-2/AM (Invitrogen) and incubated at 37 °C for 30 min in HBSS (supplemented with $Ca^{2+}/Mg^{2+}$, 1% fetal bovine serum, and 10 μg/ml antibiotics/antimycotics). After HBSS wash, cells were incubated for an additional 30 mins at 37 °C in HBSS for esterification of Fura-2/AM and transferred to the recording chamber of an inverted fluorescent microscope (Axiovert 200; Zeiss) with 20× objective lens and MetaFluor software (Molecular Device). $Ca^{2+}$ dependent fluorescence signals were obtained by using excitation at 340 nm and 380 nm, and ratioing the emission fluorescence intensities detected at 500–510 nm. The data of 1 min (60 cycles) were collected for calculation. From this ratio, the level of intracellular $Ca^{2+}$ can be estimated.

**Preparation of RNA and RNA-seq analysis.** Total RNA was extracted from RFP+ tumor cells (BxPC-3-RFP and SU.86.86-RFP) after mono-culture, indirect co-culture, or direct co-culture with PSCs for 24 h. On the other hand, total RNA was extracted from RFP+ fibroblasts after mono-culture, indirect co-culture, or direct co-culture with tumor cells (BxPC-3, SU.86.86, MIA PaCa-2, and PANC-1) for 24 h. For indirect co-culture, tumor cells and fibroblasts were separated by 0.4 μm cell culture insert (Corning). Fluorescence-activated cell sorting (FACS) was performed to collect RFP+ tumor cells or RFP+ fibroblasts after 24 h of mono- or co-culture conditions. Total RNA was subjected to cDNA synthesis and NGS library construction using the Universal Plus mRNA-Seq (Tecan Genomics, San Carlos, CA). The quality and the average length of sequence library for each sample was assessed using Bioanalyzer (Agilent Technologies, Santa Clara, CA, USA) and either the DNA 1000 kit. The indexed samples were pooled equimolarly and sequenced on Illumina NovaSeq 6000 (150 base, paired-end reads) (Illumina, San Diego, CA, USA). The quantification of raw reads was processed using CLC Genomics Workbench v.10 software. Adaptor sequences and base with low quality or ambiguous were trimmed. The quality screened reads were mapped to human (HG38) genome using CLC Genomics Workbench. The mapping parameters were the following: mismatch cost 2, insertion cost 3, deletion cost 3, length fraction of 0.5 and similarity fraction of 0.8. The expression values were calculated as FPKM (Fragments Per Kilobases per Million). The differential gene expression between two or more condition based on the fold change of FPKM value. The genes with 2-fold change were further analyzed. RNA-seq data that support the findings of this study have been deposited in the Sequence Read Archive (SRA) under accession codes from SRR16629553 to SRR16629567 within NCBI BioProject PRJNA775820.

**Quantitative real-time PCR.** Total RNAs were extracted with TRIzol® reagent (Life Technologies), and reversely transcribed with Transcriptor first strand cDNA synthesis kit (Roche). To quantify specific gene expression, the quantitative real-time RT–PCR was performed using KAPA SYBR FAST qPCR Kit (KAPA Biosystems) as manufacturer's instruction and analyzed on a Step One Plus Real-Time PCR system (Applied Biosystems, Life Technologies). Glyceraldehyde 3-phosphate dehydrogenase (GAPDH) was used as an internal control for gene expression. All primers were listed in Supplemental Table 1.

**Western blotting.** Western blotting was performed as previous reported[70]. Briefly, equal molarity of protein extracts was loaded and separated in an SDS–PAGE, and transferred to a PVDF membrane. Immunoblot analysis was performed with overnight incubation of anti-vimentin (1:1000; 5741, Cell Signaling), anti-claudin1 (1:1000; 4933, Cell Signaling), anti-Snail (1:1000; 3879, Cell Signaling), anti-Twist1 (1:1000; 711565, Invitrogen), anti-ZEB1 (1:1000; 3396, Cell Signaling), anti-ZEB2 (1:1000; 97885, Cell Signaling), anti-ATP1A1 (1:10000; 14418-1-AP, Proteintech), anti-ATP1A2 (1:1000; 16836-1-AP, Proteintech), anti-ATP1A3 (1:1000; 28030-1-AP, Proteintech), anti-EMC1 (1:1000; GTX119884, Genetex), anti-ITGB1 (1:1000; 12594-1-AP, Proteintech), anti-αSMA (1:1000; ab7817, abcam), anti-collagen1 (1:1000; ab34710, abcam), anti-Flag (1:1000; F3165, Sigma), anti-p-p65 (S536) (1:1000; 3033, Cell Signaling), and anti-p65 (1:1000; 3034, Cell Signaling). anti-GAPDH (1:10000; GTX627408, Genetex) was used as loading controls for total cell

lysates or cytosolic fraction. anti-flotillin1 (1:1000; ab133497, abcam) was used as loading control for membrane fraction.

**Lentiviral shRNA and cDNA for Infection.** Respective ATP1A1, ATP1A2, ATP1A3, EMC1, and ITGB1 depletion was performed by lentiviral transduction to introduce short hairpin RNAs (shRNA). The pLKOpuro-shATP1A1 (TRCN0000429060, sequence: 5′-CAAGCCCTTGTGATTCGAAAT-3′; TRCN0000425443, sequence: 5′-ATACCTTAACCAGTAACATTC-3′), pLKOpuro-shATP1A2 (TRCN0000043261, sequence: 5′-GATTCCTTTCAACTC-TACCAA-3′), pLKOpuro-shATP1A3 (TRCN0000072785, sequence: 5′-CGACTGTGATGACGTGAACTT-3′), pLKOpuro-shEMC1 (TRCN0000072440, sequence: 5′-GCAGAGCGATTCATCAACTAT-3′; TRCN0000072441, sequence: CGCAGAGCGATTCATCAACTA), pLKOpuro-shITGB1 (TRCN0000275133, sequence: 5′-TAGGTAGCTTTAGGGCAATAT-3′) vectors were obtained from National RNAi Core Facility (Taipei, Taiwan). The negative control was shRNA against β-galactosidase (shLacZ).

The lentiviral expression vectors for labeling cells with GFP/Luciferase and RFP were *pHAGE PGK-GFP-IRES-LUC*-W (Addgene), 7TFC (Addgene), and pLAS5w.Pbsd-L-tRFP-C (National RNAi Core Facility, Taipei, Taiwan). pLenti-ATP1A1-GFP and pLenti-ATP1A1-RFP were constructed by insertion of cDNA of ATP1A1 (NM_000701.7) at Sgf1 site and MluI site of pLenti-C-mGFP-P2A-Puro (Origene) and pLenti-C-mRFP-P2A-Puro (Origene) vectors.

Lentiviral shRNA and cDNA were packaged in HEK-293T cells. For lentivirus production, HEK-293T cells were co-transfected with lentiviral vector, packaging plasmid psPAX2, and envelope plasmid pMD2G. Virus-containing supernatant was collected 48 h post transfection. Cells were infected with lentivirus and then selected with 1–2 μg/ml puromycin.

**LC–MS/MS analysis.** Samples were detected by LC-ESI-MS on an Orbitrap Fusion mass spectrometer (Thermo Fisher Scientific, San Jose, CA) equipped with EASY-nLC 1200 system (Thermo, San Jose, CA, US) and EASY-spray source (Thermo, San Jose, CA, US). The digestion solution was injected (5 μl) at 1 μl/min flow rate on to easy column (C18, 0.075 mm × 150 mm, ID 3 μm; Thermo Scientific) Chromatographic separation was using 0.1% formic acid in water as mobile phase A and 0.1% formic acid in 80% acetonitrile as mobile phase B operated at 300 nl/min flow rate. Briefly, the gradient employed was 2% buffer B at 2 min to 40% buffer B at 40 min. Full-scan MS condition: mass range m/z 375-1800 (AGC target 5E5) with lock mass, resolution 60,000 at m/z 200, and maximum injection time of 50 ms. The MS/MS was run in top speed mode with 3 s cycles with CID for protein id, while the dynamic exclusion duration was set to 60 s with a 10-ppm tolerance around the selected precursor and its isotopes. Electrospray voltage was maintained at 1.8 kV and capillary temperature was set at 275 °C. Raw file processed by Maxquant (Version 1.6.14). MSMS spectra were searched against the uniprot Human database and enzyme digest by trypsin with 2 miss cleavage. The parameter such as MS tolerance of 20ppm for first search and main search 6 ppm. Fragment tolerance was 0.5 Da (IT). Variable modification of oxidation (M) and acetylation (protein N-terminal) and fixed modification of carbamidomethyl (C) as search parameters. The mass spectrometry proteomics data have been deposited to the ProteomeXchange Consortium via the PRIDE partner repository with the dataset identifier PXD029467 and 10.6019/PXD029467".

**Animal protocols.** All experiments using animals were approved by the Institutional Animal Care and Utilization Committee (IACUC) of Academia Sinica, Taipei, Taiwan (IACUC#14-05-709 and #20-11-1562). Mice were maintained in a SPF (specific pathogen-free) animal facility at 20 ± 2 °C and 40–60% humidity with a 12/12 h light/dark cycle and had free access to water and standard laboratory chow diet. The maximal tumor burden in our experiments does not exceed $0.3 \times 0.3 \times 0.3$ cm$^3$ under the permission of IACUC.

**Orthotopic injection.** NOD/SCID/IL2Rγ$^{null}$ (NSG) and C57BL/6 mice were anesthetized by using isoflurane. Local shaving and disinfection were performed around the left upper quadrant, and the abdominal cavity was opened by a 0.5–1 cm longitudinal incision. The spleen was lifted and the pancreatic tail was identified. Tumor cells and mixed tumor cells/fibroblasts in pre-diluted Matrigel (10 μl Matrigel + 10 μl PBS) were then slowly delivered to the pancreas using a 0.3 ml insulin syringe with 29 G needle. The pancreas was placed back, the muscle layer was closed with 6–0 absorbable surgical suture, and the skin was closed with skin staples. NSAID drug meloxicam subcutaneous injection was used for post-operative analgesia.

**Liver colonization assay.** Pancreatic tumor cells-labeled with GFP/Luciferase and fibroblasts were treated with Accutase (Merck Millipore), a cell detachment solution. A total of 10$^5$ tumor cells and $2 \times 10^5$ fibroblasts were co-incubated in 10–20 ul DMEM/F12 medium (supplemented with 10% fetal calf serum and 10 μg/ml antibiotics/antimycotics) for 30 min at room temperature. After co-incubation, tumor cell/fibroblast clusters were formed and filled in an insulin syringe within 100 μl HBSS (with $Ca^{2+}$ and $Mg^{2+}$). Before injection, insulin syringes were placed vertically to allow cells to go into the spleen first, and then HBSS flushed the cells to the liver. NOD/SCID/IL2Rγ$^{null}$ (NSG) and C57BL/6 mice were anesthetized by

using isoflurane. Local shaving and disinfection were performed around the left upper quadrant, and the abdominal cavity was opened by a 0.5–1 cm longitudinal incision. The spleen was identified. Tumor cells alone or pre-mixed with fibroblasts are slowly delivered to the spleen using a 0.3 ml insulin syringe with 29 G needle. The spleen became white upon injection and was placed back later. After injection, the muscle layer of the abdominal cavity was closed with 6-0 absorbable surgical suture, and the skin was closed with skin staples.

**Subcutaneous injection.** NOD/SCID/IL2Rγ$^{null}$ (NSG) and C57BL/6 mice were anesthetized by using isoflurane. Local shaving and disinfection were performed around the left and right flanks. Tumor cells and mixed tumor cells/fibroblasts in pre-diluted Matrigel (10 μl Matrigel + 10 μl PBS) were delivered to subcutaneous site using a 0.3 ml insulin syringe with 29 G needle.

**Statistics and reproducibility.** To analyze the effect of tumor-adjacent αSMA and INHBA on the prognosis of pancreatic cancer patients, Kaplan-Meier curves were plotted and log-rank tests were used to evaluate progression-free survival (time from surgical resection to local recurrent or distant metastasis) or overall survival (time from surgical resection to death). Student's $t$-test and Pearson's correlation coefficient were used for testing the difference and correlation between variables. The statistical analysis was performed using MedCalc 11.5.1.0 (MedCalc Software, Belgium) and Prism 7 (GraphPad Software, USA). Biological replicates were performed in all experiments except LC–MS/MS analysis and RNA-seq analysis, which are of exploratory character.

**Reporting summary.** Further information on research design is available in the Nature Research Reporting Summary linked to this article.

## Data availability

The mass spectrometry proteomics data have been deposited to the ProteomeXchange Consortium via the PRIDE partner repository with the dataset identifier PXD029467 and 10.6019/PXD029467". RNA-seq data that support the findings of this study have been deposited in the Sequence Read Archive (SRA) under accession codes from SRR16629553 to SRR16629567 within NCBI BioProject PRJNA775820. All data that support the findings of this study are provided within the article, supplementary information, and source data.

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

## Acknowledgements

We thank Dr. Che-Ming Hu, Dr. Jung-Chen Lin, and Dr. Zih-Syun Fang for their support in intracellular hydrogelation experiments, Dr. Pang-Hung Hsu for the intellectual input for designing cross-link experiments, Dr. Chein-Hung Chen for performing mass spectrometry experiment, Dr. Hsuan-Yu Peng, Huei-Fang Wu, and Yi-Ling Ko for assisting in the experiment, and Dr. Alex Ball for critical reading and suggestions. We thank GRC Mass Core Facility at Academia Sinica in Taiwan for mass analysis, the National RNAi Core Facility at Academia Sinica in Taiwan for providing shRNA plasmids, Electrophysiology and Calcium Imaging Core Facility of IMB at Academia Sinica in Taiwan for calcium imaging service, and GRC/AS Animal Core Facility at Academia Sinica in Taiwan for providing animal support. This research work was supported by funds from Academia Sinica and the Ministry of Science and Technology, Taiwan (AS-SUMMIT-109 to W.-H.L., AS-KPQ-109-BioMed to W.-H.L., AS-GC-110-MD03 to C.-M.H, MOST 109-0210-01-18-02 to W.-H.L., MOST 109-2326-B-001-019 to W.-H.L., MOST 110-2326-B-001-012 to W.-H.L.); and the higher education sprout project by the Ministry of Education, Taiwan via the "Drug Development Center of China Medical University" W.-H.L.

## Author contributions

Y.-I. C. and W.-H. L. designed the experiments. Y.-I. C., C.-C. C., and M.-F. H. conducted experiments and interpreted data. Y.-M. J., Y.-W. T., M.-C. C., and Y.-T. C. provided clinical samples. Y.-I. C., W.-H. L., and C.-M. H. wrote and completed the manuscript. W.-H. L. and C.-M. H. coordinated and supervised the entire project.

## Competing interests

The authors declare no competing interests.
