## [Peer Review File · Nature Communications]

Homophilic ATP1A1 binding induces activin A Secretion to promote EMT of tumor cells and myofibroblast activationEditorial Note: Parts of this Peer Review File have been redacted as indicated to remove third-party material where no permission to publish could be obtained.

REVIEWER COMMENTS

Reviewer #1 (Remarks to the Author):

This article claims to demonstrate a novel association between juxta-tumoral myofibroblasts and EMT of tumor cells in PDAC. The article is a comprehensive and well written. The authors convincingly demonstrate a role for CAF derived Activin A in this process and identifies homophilic ATP1A1 binding as a trigger for Activin A secretion.

But the authors also claim that it is myofibroblasts that are responsible for driving this EMT process. Based on the data provided, I am not fully convinced that this is the case. It has to be shown that the cell-based experiments in figure 2 and 3 are involving myofibroblasts, and not other CAF subtypes. The cells used in the experiments are PSCs, and they can in co-culture with cancer cells convert into a myofibroblastic phenotype, but also other phenotypes, such as an inflammatory phenotype. These phenotypes are mutually exclusive. For instance, in figure 2f and 2i, PSCs are shown to form aggregates with cancer cells. But are the PSCs in the cell aggregates myofibroblasts? This has to be shown. Furthermore, in figure 3, RNA-seq was performed on cancer cells, and an EMT phenotype could be detected. But RNA-seq should also be performed on the fibroblast, to confirm that these juxta-tumoral PSCs have converted to a myofibroblastic phenotype (when compared to monocultures of PSCs).

Interestingly, this article contradicts a number of previous studies which have suggested juxta-tumoral myofibroblasts may have a tumor-suppressive role. It would also be of merit for this article to acknowledge the lack of consensus in the literature with respect to the role of CAF subpopulations orchestrating tumor promoting or restraining roles and discuss how the current findings contradict previous opposing publications.

Minor concerns:

Abstract:

No comments

Introduction:

The introduction is well written and motivates the study well. The TME is often represented as a universally tumor promoting niche. There is a significant number of studies suggesting stromal components may also orchestrate tumor restraining roles. It would be of merit to briefly acknowledge that there is not consensus in the literature of the stroma being unequivocally tumor promoting.

Results:

In figure 1i and 1j the correlations of aSMA expression within 10 um to cancer cells and outcome are shown. A similar analysis should be done for aSMA expression more distant (10-20 um) to cancer cells. This would make it clear that it is the juxta-tumoral localization of aSMA that is important and associated to survival, not aSMA expression in general.

Page 5 line 5: "biomarker for a subset of activated fibroblasts (termed "myofibroblasts")" – aSMA is not widely considered a universal marker of activated fibroblasts but rather of myofibroblasts specifically.

Page 7 line 25: "(Supplementary Fig. 2e, f)" – there is no Supplementary Fig 2e,f.

Page 7 line 19: anti-activin A or Folistatin only showed a reduction in tumour growth in cases where PSCs are also injected. Present text implies a reduction in tumour growth when compounds are added irrespective of PSC status. Consistently, aSMA and collagen I were also increased in fibroblasts in direct contact with tumor cells (Fig. 4o, p; Supplementary Fig. 3d) – Supplementary Figure 3d quantifies contractile forces in culture conditions which is not mentioned.

Page 9 line 12: Supplementary fig 4i does not exist (neither does h for that matter, although I

don't see this referred to in the text). It seems likely that the author intends to refer to supplementary figure 5i in this case.

Figure 7i: It has previously been reported (Biffi G, Oni TE, Spielman B, et al. IL1-induced Jak/STAT signaling is antagonized by TGF β to shape CAF heterogeneity in pancreatic ductal adenocarcinoma. *Cancer Discov.* 2019;9(2):282-301. doi:10.1158/2159-8290.CD-18-0710) that indirect coculture or treatment with conditioned media is able to induce NF κ B signalling in PSCs and that this is associated with inflammatory CAFs (iCAFs) which are located more distally to tumour cells. There isn't a comparison of PSCs in direct contact with tumour cells. It would help to compare P65 phosphorylation between monocultured PSCs, indirect and direct coculture to determine whether NF κ B signalling is elevated in indirect coculture compared to monoculture which would be consistent with previous findings.

The above-mentioned publication also finds that TGF β secreted by tumor cells downregulates inflammatory signaling pathways such as NF κ B in myofibroblasts in close contact with tumour cells. As this is an opposing finding to the current study it merits discussion.

Discussion:

A landmark study (Özdemir BC, Pentcheva-Hoang T, Carstens JL, et al. Depletion of carcinoma-associated fibroblasts and fibrosis induces immunosuppression and accelerates pancreas cancer with reduced survival. *Cancer Cell.* 2014;25(6):719-734. doi:10.1016/j.ccr.2014.04.005) demonstrated that depletion of α SMA+ fibroblasts gave rise to undifferentiated invasive tumors. The present study demonstrates opposing findings and should discuss how these contradicting conclusions may have arisen.

Materials and methods:

No specific comments.

Other:

Page 9 line 4/5: minor formatting issue of no line break between paragraphs.

Page 13 line 14/15: minor formatting issue of no line break between paragraphs.

Page 15 line 13/14: minor formatting issue of no line break between paragraphs.

Page 20 line 15/16: minor formatting issue of no line break between paragraphs.

Page 22 line 6/7: minor formatting issue of no line break between paragraphs.

Reviewer #2 (Remarks to the Author):

The manuscript "Homophilic ATP1A binding induces Activin A secretion to promote epithelial-mesenchymal transition of tumor cells and myofibroblast activation" by Chen et al. describes the direct contact between myofibroblasts and pancreatic tumor cells mediated by ATP1A as an important step in the induction of Activin A secretion and EMT.

Strength:

While the idea that myofibroblasts or cancer-associated fibroblasts interact with tumor cells and can induce EMT resulting in the progression and aggressiveness, e.g., migration and invasion of PDAC cells, has been the topic of many investigations, the mechanistic focus on Activin A secretion as a mediator is novel to this comprehensive study. The study is extremely logically built and addresses additional questions to what stimulates Activin secretion in the first place and what are the consequences of this interaction, e.g., downstream mediators such as NF- κ B.

Weaknesses:

The authors establish that the distance between α SMA+ myofibroblasts and tumor cells is shorter indicating more potential interaction than that of tumor with the general vimentin+ population. It seems that the approach in Figure 2 to identify the different cell populations within a tumor cluster or a microemboli from peripheral blood should still address if only α SMA+ or also vimentin+ cells are part of these clusters. Aside from staining only for α SMA, vimentin should be added as a 'control' to demonstrate the enrichment.

Are mouse pancreatic stellate cells α SMA+? Would it be important to show that the stellate cells are not just vimentin+?

Statistical analysis:
Adequate

Reviewer #3 (Remarks to the Author):

Chen et al. describe PDAC-CAF interactions driving EMT and the presence of circulating tumor microemboli (CTM) with PDAC-CAF heterocellular aggregates. This is driven by ATP1A1 in tumor cells that alter fibroblasts to secrete activin A which is important for invasion. Alpha-SMA positive CAFs are found to surround PDAC tumor cells. These CAFs in co-culture drives EMT states in PDAC tumor cells in vitro. Using a cytokine array, the authors identify activin A as a potential driver of CAF mediated PDAC EMT. Recombinant activin A induces EMT and depleting activin A antibody reduces EMT as expected. This is translated in vivo. Using surface proteomics and functional knockdown in PDAC cells, the authors find ATP1A1 as the mediator of PDAC driven activin A CAF secretion. This ATP1A1 homophilic interaction drives intracellular Ca and NF κ B signaling in CAFs that leads to Activin A secretion. Overall an impressive amount of work with some interesting findings between PDAC and CAF cells.

Comments:

- 1) Abstract – grammar of sentence “Mechanistically, the overexpressed ATP1A1...” is awkward
- 2) Introduction – A lot of reviews are cited but should cite primary publications and focus on PDAC EMT papers instead of ones in other cancers that are cited. For example, consider citation of Hwang Cancer Research 2008; Erdogan et al. JCB 2017 and Ligorio et al. Cell 2019 given prior work showing CAFs induce PDAC EMT heterogeneity
- 3) Figure 2c – how do you know that the alphaSMA positive cell is not an PDAC tumor cell that has undergone EMT? (i.e. if you stain mouse PDAC tumors with alpha-SMA IHC or ISH do you see tumor cells that are alpha-SMA positive. Based on prior scRNA-seq in PDAC, alpha-SMA is not CAF specific and is expressed in many PDAC cell lines)
- 4) Figure 5d-g – if you look at the INHBA positive CAF tumor samples, do you see higher PDAC EMT tumor cells (one could do RNA-ISH for FN1 and KRTs to see if the number of FN1+ KRT+ cells is higher with INHBA+ CAFs)
- 5) Does ATP1A1 in PDAC affect CAF polarization (myCAF vs iCAF)? Alpha-SMA has been associated with myCAF, but does the ATP1A1 homophilic interaction change this myCAF state to be more iCAF like to secrete Activin A or do they become even more myCAF like as described by the Tuveson group? (Biffi et al. Cancer Discovery 2018; Elyada Cancer Discovery 2019). There is emerging data indicating that the myCAFs and iCAFs are on a spectrum similar to E vs M state in tumor cells.

Detailed point-by-point response to reviewers' comments

Reviewer #1 (Remarks to the Author):

Summary

This article claims to demonstrate a novel association between juxta-tumoral myofibroblasts and EMT of tumor cells in PDAC. The article is a comprehensive and well written. The authors convincingly demonstrate a role for CAF derived Activin A in this process and identifies homophilic ATP1A1 binding as a trigger for Activin A secretion.

Response: We appreciate the reviewer's positive comments.

Comments:

Comment 1. But the authors also claim that it is myofibroblasts that are responsible for driving this EMT process. Based on the data provided, I am not fully convinced that this is the case. It has to be shown that the cell-based experiments in figure 2 and 3 are involving myofibroblasts, and not other CAF subtypes. The cells used in the experiments are PSCs, and they can in co-culture with cancer cells convert into a myofibroblastic phenotype, but also other phenotypes, such as an inflammatory phenotype. These phenotypes are mutually exclusive. For instance, in figure 2f and 2i, PSCs are shown to form aggregates with cancer cells. But are the PSCs in the cell aggregates myofibroblasts? This has to be shown.

Response: We appreciate this suggestion. We have performed IF staining with anti- α SMA antibodies in PSCs that form aggregates with tumor cells. As shown in R. Figure 1, fibroblasts in 3D-based tumor-fibroblast aggregates (R. Figure 1a), tumor-fibroblast spheroids (R. Figure 1b), and spheroid invasion front (R. Figure 1c) express α SMA, a marker of myofibroblasts. These results have been incorporated in Supplemental Fig. 1f-h in Page 6 line 7-8.

R. Figure 1. PSCs that form aggregates with tumor cells express α SMA. Immunofluorescence staining of α SMA in (a) fibroblasts (mPSC-RFP and PSC-GFP) and cancer cells (PdKP53-GFP and BxPC-3-RFP) in 3D-based tumor-fibroblast aggregates, (b) fibroblasts (PSC-GFP) and cancer cells (BxPC-3-RFP and SU.86.86-RFP) in tumor-fibroblast spheroids, and (c) fibroblasts (PSC-GFP) and cancer cells (BxPC-3-RFP) in spheroid invasion front measured by confocal microscope. Scale bar, 50 μ m.

Comment 2. Furthermore, in figure 3, RNA-seq was performed on cancer cells, and an EMT phenotype could be detected. But RNA-seq should also be performed on the fibroblast, to confirm that these juxta-tumoral PSCs have converted to a myofibroblastic phenotype (when compared to monocultures of PSCs).

Response: As shown in R. Figure 2a and 2b, we compared the gene expression profiles of PSCs between mono- or direct- co-cultured conditions with BxPC-3 cells. We identified 5325 up-regulated (≥ 2 -fold) genes and the top canonical pathways associated with these genes were related to fibroblast activation (R. Figure 2a). Among the genes related to fibroblast activation, genes known to be up-regulated in myofibroblasts (red mark) including ACTA2 (gene name of α SMA), Collagen, and TGF β were noted (R. Figure 2b). Genes related to iCAF (green mark) including IL6 and IL11 were also identified (R. Figure 2b). Based on the biomarkers of myCAF and iCAF published by Öhlund's and Biffi^{1, 2}, we further examined their expression levels under our experimental condition. As shown in R. Figure 2c and 2d, the fibroblasts direct contact with tumor cells up-regulate not only all myCAF marker genes, but also some iCAF (IL11, IL6, and PI16) marker genes. These data suggest that the myCAF may also express some biomarker genes of iCAF. However, the other possibility is only a small portion of CAFs express iCAF biomarker genes, which are not expressed in myCAF. Since this aspect is not the

key point for this investigation, we believe that further detailed study is needed to clarify this issue.

R. Figure 2. Fibroblasts direct contact with tumor cells up-regulate all myCAF marker genes. (a) Top 5 canonical pathways with the largest -log (p-value). **(b)** Up-regulated genes in pathways related to fibroblast activation. myCAF genes were marked by red color and iCAF genes were marked by green color. **(c and d)** Quantitative real-time PCR analyses of genes related to myCAF (ACTA2, COL1A1, and TGFB1) and iCAF (IL11, IL6, LIF, CXCL12, and PI16) in fibroblasts (PSC-GFP) that were mono-cultured, indirect co-cultured (ID), and direct co-cultured (D) with tumor cells (BxPC-3). Fluorescence activated cell sorting (FACS) was performed to collect GFP⁺ fibroblasts after 24 hrs of co-culture. Values were presented as mean ± SD (n = 3).

Reference:

1. Öhlund D, *et al.* Distinct populations of inflammatory fibroblasts and myofibroblasts in pancreatic cancer. *J Exp Med* **214**, 579-596 (2017).
2. Biffi G, *et al.* IL1-Induced JAK/STAT Signaling Is Antagonized by TGFβ to Shape CAF Heterogeneity in Pancreatic Ductal Adenocarcinoma. *Cancer Discovery* **9**, 282 (2019).

Comment 3. Interestingly, this article contradicts a number of previous studies which have suggested juxta-tumoral myofibroblasts may have a tumor-suppressive role. It would also be of merit for this article to acknowledge the lack of consensus in the literature with respect to the role of CAF subpopulations orchestrating tumor promoting or restraining roles and discuss how the current findings contradict previous opposing publications.

Response: Indeed, there are some contradictions with this finding in the literatures. The notion that juxta-tumoral myofibroblasts have tumor-suppressive roles was concluded from two studies. Özdemir BC *et al.*³ demonstrated that depletion of proliferating α SMA⁺ myofibroblasts in a α SMA-tk (thymidine kinase) transgenic mice with ganciclovir resulted in undifferentiated PDAC and decreased survival of the mice, suggesting that α SMA⁺ fibroblasts may restrain tumor progression. The other study from Öhlund D. *et al.* showed that a distinct population of α SMA⁺ myofibroblasts¹ was localized at juxta-tumor. These two findings hint that juxta-tumoral α SMA⁺ myofibroblasts may have tumor-suppressive role. However, juxta-tumoral myofibroblasts possess stemness properties⁴ and are not highly proliferative. In fact, they are preserved upon depletion of α SMA⁺ cells by ganciclovir as shown in R. Figure 3., which were from their original paper (Özdemir BC *et al.*³). Thus, their conclusion may not be decisive. We shall add this point at the discussion section.

[REDACTED]

R. Figure 3. Figure 3A from the main figure of Özdemir BC. *et al.*³ Representative micrographs (scale bar represents 50 μ m) of collagen I and α SMA.

Reference:

1. Öhlund D, *et al.* Distinct populations of inflammatory fibroblasts and myofibroblasts in pancreatic cancer. *J Exp Med* **214**, 579-596 (2017).

3. Özdemir BC, *et al.* Depletion of carcinoma-associated fibroblasts and fibrosis induces immunosuppression and accelerates pancreas cancer with reduced survival. *Cancer Cell* **25**, 719-734 (2014).
4. Su S, *et al.* CD10(+)GPR77(+) Cancer-Associated Fibroblasts Promote Cancer Formation and Chemoresistance by Sustaining Cancer Stemness. *Cell* **172**, 841-856.e816 (2018).

Minor comments:

1. Abstract: No comments.

Response: Thank you.

2. Introduction: The introduction is well written and motivates the study well. The TME is often represented as a universally tumor promoting niche. There is a significant number of studies suggesting stromal components may also orchestrate tumor restraining roles. It would be of merit to briefly acknowledge that there is not consensus in the literature of the stroma being unequivocally tumor promoting.

Response: Thank you for the comments. We have modified paragraph 2 and 3 to clarify the point. In paragraph 2, we added “In the beginning of tumorigenesis when phenotypic normal epithelial cells encumber with mutations, differentiated cues derived from surrounding stromal cells would restrain malignant growth.” in Page 3 line 13-14. In paragraph 3, we added “tumor-promoting or restraining functions” in Page 3 line 27-28.

3. Results: In figure 1i and 1j the correlations of α SMA expression within 10 μ m to cancer cells and outcome are shown. A similar analysis should be done for α SMA expression more distant (10-20 μ m) to cancer cells. This would make it clear that it is the juxta-tumoral localization of α SMA that is important and associated to survival, not α SMA expression in genera.

Response: Kaplan-Meier survival curve of α SMA expression levels in 10~20 μ m regions was not correlated to patient outcome as shown in R. Figure 4. These results have been incorporated in Fig. 1j and 1l in Page 5 line 21-22.

R. Figure 4. α SMA expression levels in 10~20 μ m regions were not associated to patient survival. Kaplan-Meier analysis of (a) progression-free survival and (b) overall survival according to α SMA expression levels in regions within 10~20 μ m distance from tumor.

4. Page 5 line 5: “biomarker for a subset of activated fibroblasts (termed “myofibroblasts”)” – α SMA is not widely considered a universal marker of activated fibroblasts but rather of myofibroblasts specifically.

Response: Thank you for the correction. We rewrote the sentence as “biomarker for myofibroblasts” in Page 5 line 5.

5. Page 7 line 25: “(Supplementary Fig. 2e, f)” – there is no Supplementary Fig 2e,f.

Response: It should be “Supplementary Fig. 2c, d” instead of “Supplementary Fig. 2e, f”. We corrected the error in Page 7 line 25.

6. Page 7 line 19: anti-activin A or Folistatin only showed a reduction in tumour growth in cases where PSCs are also injected. Present text implies a reduction in tumour growth when compounds are added irrespective of PSC status. Consistently, α SMA and collagen I were also increased in fibroblasts in direct contact with tumor cells (Fig. 4o, p; Supplementary Fig. 3d)

Response: Thank you. We added “only in cases where PSCs were also injected in Page 7 line 19.

7. Supplementary Figure 3d quantifies contractile forces in culture conditions which is not mentioned.

Response: Thank you. We have mentioned the result in Page 7 line 28.

8. Page 9 line 12: Supplementary fig 4i does not exist (neither does that matter, although I don't see this referred to in the text). It seems likely that the author intends to refer to supplementary figure 5i in this case.

Response: Thank you. We have changed Supplementary Fig. 4i to 5i in Page 9 line 14.

9. Figure 7i: It has previously been reported (Biffi G, Oni TE, Spielman B, et al. I11-induced Jak/STAT signaling is antagonized by TGF β to shape CAF heterogeneity in pancreatic ductal adenocarcinoma. *Cancer Discov.* 2019;9(2):282-301. doi:10.1158/2159-8290.CD-18-0710) that indirect coculture or treatment with conditioned media is able to induce NF κ B signaling in PSCs and that this is associated with inflammatory CAFs (iCAFs) which are located more distally to tumour cells. There isn't a comparison of PSCs in direct contact with tumour cells. It would help to compare P65 phosphorylation between monocultured PSCs, indirect and direct coculture to determine whether NF κ B signaling is elevated in indirect coculture compared to monoculture which would be consistent with previous findings.

Response: Thank you for the suggestion. Comparison of P65 phosphorylation among PSCs in mono-, indirect- and direct- co-cultured conditions is shown in R. Figure 5. NF- κ B signaling in PSCs is up-regulated in indirect co-cultured condition and further up-regulated in direct co-cultured condition. Therefore, our result is consistent with the previous finding. This result is incorporated in Supplementary Fig. 7d in Page 11 line 5-7.

R. Figure 5. NF- κ B signaling in PSCs is up-regulated in indirect co-cultured condition and further up-regulated in direct co-cultured condition. Western blotting analysis of proteins harvested from mono-, indirect-, and direct- co-cultured fibroblast-GFP and BxPC-3 cells. Fluorescence activated cell sorting (FACS) was performed to collect GFP⁺ and GFP⁻ cells after 48 hours of mono-, indirect-, or direct- co-cultured conditions.

10. The above-mentioned publication also finds that TGF β secreted by tumor cells downregulates inflammatory signaling pathways such as NF- κ B in myofibroblasts in close contact with tumor cells. As this is an opposing finding to the current study it merits discussion.

Response: According to the results from Biffi G. *et al.*, TGF β downregulates IL1R1 expression in PSC to prevent the activation of cytokines (IL1a, IL6, CXCL1, and CSF3) downstream of JAK/STAT signaling that lead to iCAF formation². However, their results did not address whether NF- κ B pathway was downregulated by TGF β in myofibroblasts in close contact with tumor cells.

To prove that NF- κ B signaling in juxta-tumoral myofibroblasts is activated by ATP1A1 interaction, we knockdown ATP1A1 in tumor cells to block ATP1A1 interactions between BxPC-3 cells and PSCs. As shown in R. Figure 6, the activation of NF- κ B signaling in PSCs was decreased when ATP1A1 interactions were abolished. This result is now incorporated in the Fig. 7k in Page 11 line 5-7.

R. Figure 6. NF- κ B signaling in PSCs is activated by ATP1A1 interactions to tumor cells. Western blotting analysis of proteins harvested from indirect- and direct- co-cultured PSC cells with BxPC-3-RFP stably expressing lentiviral-based LacZ^{shRNA} and ATP1A1^{shRNA Clone#1}. Fluorescence activated cell sorting (FACS) was performed to collect RFP⁻ cells after 48 hours of indirect (ID)- or direct (D)- co-cultured conditions.

Reference:

2. Biffi G, *et al.* IL1-Induced JAK/STAT Signaling Is Antagonized by TGF β to Shape CAF Heterogeneity in Pancreatic Ductal Adenocarcinoma. *Cancer Discovery* **9**, 282 (2019).

11. Discussion: A landmark study (Özdemir BC, Pentcheva-Hoang T, Carstens JL, *et al.* Depletion of carcinoma-associated fibroblasts and fibrosis induces immunosuppression and accelerates pancreas cancer with reduced survival. *Cancer Cell*. 2014;25(6):719-734. doi:10.1016/j.ccr.2014.04.005) demonstrated that depletion of α SMA⁺ fibroblasts gave rise to undifferentiated invasive tumors. The present study demonstrates opposing findings and should discuss how these contradicting conclusions may have arisen.

Response: As explained in Comment 3, we add this point in our discussion to reconcile these findings.

12. Materials and methods: No specific comments.

Response: Thank you.

13. Other: Page 9 line 4/5: minor formatting issue of no line break between paragraphs. Page 13 line 14/15: minor formatting issue of no line break between paragraphs. Page 15 line 13/14: minor formatting issue of no line break between paragraphs. Page 20 line 15/16: minor formatting issue of no line break between paragraphs. Page 22 line 6/7: minor formatting issue of no line break between paragraphs.

Response: Thank you for the correction.

Reviewer #2 (Remarks to the Author):

Summary

The manuscript “Homophilic ATP1A binding induces Activin A secretion to promote epithelial-mesenchymal transition of tumor cells and myofibroblast activation” by Chen et al. describes the direct contact between myofibroblasts and pancreatic tumor cells mediated by ATP1A as an important step in the induction of Activin A secretion and EMT.

Response: We appreciate the reviewer’s positive comments.

Comment 1. Strength: While the idea that myofibroblasts or cancer-associated fibroblasts interact with tumor cells and can induce EMT resulting in the progression and aggressiveness, e.g., migration and invasion of PDAC cells, has been the topic of many investigations, the mechanistic focus on Activin A secretion as a mediators is novel to this comprehensive study. The study is extremely logically built and addresses additional questions to what stimulates Activin secretion in the first place and what are the consequences of this interaction, e.g., downstream mediators such as NF-KB.

Response: We appreciate the reviewer’s positive comments.

Comment 2. Weaknesses: The authors establish that the distance between aSMA+ myofibroblasts and tumor cells is shorter indicating more protential interaction than that of tumor with the general vimentin+ population. It seems that the approach in Figure 2 to identify the different cell populations within a tumor cluster or a microemboli from peripheral blood should still address if only aSMA+ or also vimentin+ cells are part of these clusters. Aside from staining only for aSMA, vimentin should be added as a ‘control’ to demonstrate the enrichment.

Response: We used vimentin to indicate all fibroblast as written in Page 5 line 9, and α SMA to indicate myofibroblasts as written in Page 5 line 5. The distance between α SMA+ myofibroblasts and tumor cells is shorter indicating a potential interaction of tumor with α SMA+/vimentin+ myofibroblasts than that of tumor with general vimentin+ fibroblasts. Therefore, fibroblasts that direct contact with tumor cells are

α SMA⁺/vimentin⁺ myofibroblasts as shown in the original Figure 1b and 1f. Furthermore, fibroblasts in EpCAM⁺/PDGFR α ⁺ cell clusters from pancreatic tumor and circulating tumor microemboli from peripheral blood express α SMA and vimentin simultaneously as shown in R. Figure 7.

Figure 1b and 1f. α SMA⁺ fibroblasts were enriched in juxta-tumoral regions, while VIM⁺ fibroblasts were evenly distributed throughout the tumor stroma. Representative IHC images of VIM/ α SMA/DAPI expression in (b) PdKP53 transgenic mice and (f) human PDAC specimens. Scale bar, 10 μ m.

R. Figure 7. α SMA⁺/VIM⁺ myofibroblasts directly contact with tumor cells. Immunostaining of α SMA and VIM expression in tumor-fibroblast clusters isolated from (a) mechanical dissociated pancreatic tumor and (b) circulating tumor microemboli of PdKP53 mice. Scale Bar, 10 μ m.

Comment 3. Are mouse pancreatic stellate cells α SMA⁺? Would it be important to show that the stellate cells are not just vimentin⁺?

Response: Yes. A portion of mouse pancreatic stellate cells are α SMA⁺, IF staining of vimentin and α SMA in mouse stellate cells are shown in R. Figure 8.

R. Figure 8. Mouse pancreatic stellate cells express α SMA and vimentin. Immunostaining of α SMA and vimentin in mouse pancreatic stellate cells. Scale Bar, 50 μ m.

Comment 4. Statistical analysis: Adequate

Response: Thank you.

Reviewer #3 (Remarks to the Author):

Summary

Chen et al. describe PDAC-CAF interactions driving EMT and the presence of circulating tumor microemboli (CTM) with PDAC-CAF heterocellular aggregates. This is driven by ATP1A1 in tumor cells that alter fibroblasts to secrete activin A which is important for invasion. Alpha-SMA positive CAFs are found to surround PDAC tumor cells. These CAFs in co-culture drives EMT states in PDAC tumor cells in vitro. Using a cytokine array, the authors identify activin A as a potential driver of CAF mediated PDAC EMT. Recombinant activin A induces EMT and depleting activin A antibody reduces EMT as expected. This is translated in vivo. Using surface proteomics and functional knockdown in PDAC cells, the authors find ATP1A1 as the mediator of PDAC driven activin A CAF secretion. This ATP1A1 homophilic interaction drives intracellular Ca and NF κ B signaling in CAFs that leads to Activin A secretion. Overall an impressive amount of work with some interesting findings between PDAC and CAF cells.

Response: We appreciate the reviewer’s positive comments.

Comment 1. Abstract – grammar of sentence “Mechanistically, the overexpressed ATP1A1...” is awkward

Response: Thank you. We have revised the sentence to “Mechanistically, ATP1A1 overexpressed in tumor cells binds to and reorganizes ATP1A1 of fibroblasts that induces calcium oscillations, NF- κ B activation, and activin A secretion.” in Page 2 line 8-9.

Comment 2. Introduction – A lot of reviews are cited but should cite primary publications and focus on PDAC EMT papers instead of ones in other cancers that are cited. For example, consider citation of Hwang Cancer Research 2008; Erdogan et al. JCB 2017 and Ligorio et al. Cell 2019 given prior work showing CAFs induce PDAC EMT heterogeneity

Response: Thank you for the suggestions. The references have been edited.

Comment 3. Figure 2c – how do you know that the alphaSMA positive cell is not an PDAC tumor cell that has undergone EMT? (i.e. if you stain mouse PDAC tumors with alpha-SMA IHC or ISH do you see tumor cells that are alpha-SMA positive. Based on prior scRNA-seq in PDAC, alpha-SMA is not CAF specific and is expressed in many PDAC cell lines)

Response: We appreciate the suggestion. The following data support that the α SMA positive cell is not a PDAC tumor cell that has undergone EMT. First, in the original Figure 2c, circulating tumor microemboli is captured from peripheral blood of *Pdx1-Cre; LSL-Kras^{G12D/+}; Trp53^{flox/flox}* (PdKP53) mice, in which α SMA positive cell is not a PDAC tumor cell that has undergone EMT stained by CK19. Second, CK19 is expressed in all tumor regions in both primary tumor and liver metastasis, while tumor-adjacent CK19- stromal cells express significantly higher α SMA levels (R. Figure 9a). This observation is similar to that of human PDAC tumor specimens which 99.5% of them express CK19⁵. Therefore, a CK19⁻/ α SMA⁺ cell is not a PDAC tumor cell that has undergone EMT. Third, we co-stained Cre with Acta2 (gene name of α SMA) in tumor specimen derived from PdKP53 mice using *in situ* hybridization. The result showed that tumor-adjacent α SMA positive fibroblasts were Cre-negative, in contrary to tumor regions (R. Figure 9b). This lineage tracing proved that tumor-adjacent α SMA positive cells is a distinct cell type different from tumor cells generated by oncogenic *Kras^{G12D}* in *Pdx1⁺* pancreatic progenitor cells.

Figure 2c. α SMA⁺ fibroblasts are CK19 negative. Representative IF images of CK19/ α SMA/CD45/DAPI expression in circulating tumor microemboli in 6 to 7 weeks old PdKP53 transgenic mice. Scale bar, 10 μ m.

R. Figure 9. CK19 is expressed in tumor cells but not tumor-adjacent α SMA⁺ cells of PdKP53 mice. (a) Representative IHC images of CK19/ α SMA/DAPI expression in the primary tumor and liver metastasis of 6 to 7 weeks old *Pdx1-Cre*; *LSL-Kras*^{G12D/+}; *Trp53*^{flox/flox} (PdKP53) transgenic mice. Scale bar, 50 μ m. (b) *In situ* hybridization (RNAscope, ACDBio) using probes for co-staining of Cre (blue color) and Acta2 (gene name of α SMA, pink color) in PdKP53 mice. Scale Bar, 50 μ m.

Reference:

5. Menz A, *et al.* Diagnostic and prognostic impact of cytokeratin 19 expression analysis in human tumors: a tissue microarray study of 13,172 tumors. *Human Pathology* **115**, 19-36 (2021).

Comment 4. Figure 5d-g – if you look at the INHBA positive CAF tumor samples, do you see higher PDAC EMT tumor cells (one could do RNA-ISH for FN1 and KRTs to see if the number of FN1+ KRT+ cells is higher with INHBA+ CAFs)

Response: Yes. To analyze the EMT expression levels of PDAC tumor cells, we stained vimentin (an EMT marker) and CK19 (a PDAC marker). We have observed that the number of vimentin⁺ CK19⁺ tumor cells is higher at the region adjacent to INHBA positive fibroblasts (INHBA⁺/ACTA2⁺) in human PDAC specimens (R. Figure 10). These results are incorporated in Fig. 5f and 5g in Page 8 line 10-12.

R. Figure 10. Enhancement of EMT at tumor regions adjacent to activin A-secreting fibroblasts. (a) Serial sections of ACTA2/INHBA expression (ACTA2, gene name of α SMA, blue color; INHBA, gene name of activin A, pink color, in situ hybridization, RNAscope) and CK19/VIM expression (brown color, IHC) in human PDAC specimens. Scale bar, 50 μ m. (b) H score of VIM expression levels in tumor cells with tumor-adjacent INHBA⁻ or INHBA⁺ CAFs (n = 30).

Comment 5. Does ATP1A1 in PDAC affect CAF polarization (myCAF vs iCAF)? Alpha-SMA has been associated with myCAF, but does the ATP1A1 homophilic interaction change this myCAF state to be more iCAF like to secrete activin A or do they become even more myCAF like as described by the Tuveson group? (Biffi *et al.* Cancer

Discovery 2018; Elyada Cancer Discovery 2019). There is emerging data indicating that the myCAFs and iCAFs are on a spectrum similar to E vs M state in tumor cells.

Response: Since ATP1A1 interactions depend on direct cell-cell contact, we compared the gene expression profiles of PSCs between mono- or direct- co-cultured conditions with BxPC-3 cells using RNA-seq data. We identified 5325 up-regulated (≥ 2 -fold) genes and the top canonical pathways associated with these genes were related to fibroblast activation (R. Figure 11a). Among the genes related to fibroblast activation, genes known to be up-regulated in myofibroblasts (red mark) including ACTA2 (gene name of α SMA), Collagen, and TGFB were noted (R. Figure 11b). Genes related to iCAF (green mark) including IL6 and IL11 were also up-regulated in direct-cocultured condition (R. Figure 11b). Based on the biomarkers published by Öhlund's and Biffi^{1,2}, we further examined their expression under our experimental condition. As shown in R. Figure 11c and 11d, the fibroblasts direct contact with tumor cells not only up-regulates all myCAF marker genes, but also some iCAF (IL11, IL6, and PI16) marker genes.

To validate whether ATP1A1 homophilic interactions is crucial for gene expression change regulated by direct contact, we compare genes related to myCAFs and iCAFs in mono-cultured and direct co-cultured PSCs with BxPC-3 cells stably expressing lentiviral-based LacZ^{shRNA} and ATP1A1^{shRNA} (R. Figure 11e and 11f). When ATP1A1 interactions were abolished, expression levels of ACTA2 (gene name of α SMA) and PI16 were down-regulated. Thus, ATP1A1 interaction modulates α SMA expression which is consistent with the finding that direct contact activates NF- κ B signaling and secretes activin A to induce α SMA expression (Fig. 8).

Based on these two sets of the data, the effect of direct contact on CAF polarization is more than the ATP1A1 interaction. Further detailed study in this regard is warranted.

R. Figure 11. Fibroblasts direct contact with tumor cells up-regulate ACTA2 and PI16 through ATP1A1 interactions. (a) Top 5 canonical pathways with the largest -log (p-value). (b) Up-regulated genes in pathways related to fibroblast activation. myCAF genes were marked by red color and iCAF genes were marked by green color. (c-f) Quantitative real-time PCR analyses of genes related to myCAF (ACTA2, COL1A1, and TGFB1) and iCAF (IL11, IL6, LIF, CXCL12, and PI16) in (c and d) mono-cultured, indirect co-cultured (ID), and direct co-cultured (D) fibroblasts (PSC-GFP) with tumor cells (BxPC-3) and (e and f) mono-cultured and direct co-cultured PSCs with BxPC-3-RFP cells stably expressing lentiviral-based LacZ^{shRNA} and ATP1A1^{shRNA} Clone#1. Fluorescence activated cell sorting (FACS) was performed to collect GFP⁺ fibroblasts after 24 hrs of co-culture. Values were presented as mean \pm SD (n = 3).

Reference:

1. Öhlund D, *et al.* Distinct populations of inflammatory fibroblasts and myofibroblasts in pancreatic cancer. *J Exp Med* **214**, 579-596 (2017).
2. Biffi G, *et al.* IL1-Induced JAK/STAT Signaling Is Antagonized by TGF β to Shape CAF Heterogeneity in Pancreatic Ductal Adenocarcinoma. *Cancer Discovery* **9**, 282 (2019).
3. Özdemir BC, *et al.* Depletion of carcinoma-associated fibroblasts and fibrosis induces immunosuppression and accelerates pancreas cancer with reduced survival. *Cancer Cell* **25**, 719-734 (2014).
4. Su S, *et al.* CD10(+)GPR77(+) Cancer-Associated Fibroblasts Promote Cancer Formation and Chemoresistance by Sustaining Cancer Stemness. *Cell* **172**, 841-856.e816 (2018).
5. Menz A, *et al.* Diagnostic and prognostic impact of cytokeratin 19 expression analysis in human tumors: a tissue microarray study of 13,172 tumors. *Human Pathology* **115**, 19-36 (2021).

REVIEWERS' COMMENTS

Reviewer #1 (Remarks to the Author):

The authors have now addressed all my previous comments and questions adequately. Important new experiments have been added. I have no further comments.

Reviewer #2 (Remarks to the Author):

My concerns have been addressed with the revised version of this manuscript.

Reviewer #3 (Remarks to the Author):

The authors have taken considerable time and effort in addressing my concerns. No further comments/questions.

Detailed point-by-point response to reviewers' comments

Reviewer #1 (Remarks to the Author):

Summary

The authors have now addressed all my precious comments and questions adequately. Important new experiments have been added. I have no further comments.

Response: Thank you.

Reviewer #2 (Remarks to the Author):

Summary

My concerns have been addressed with the revised version of this manuscript.

Response: Thank you.

Reviewer #3 (Remarks to the Author):

Summary

The authors have taken considerable time and effort in addressing my concerns. No further comments/questions.

Response: Thank you.